# Cdk5 and GSK3β inhibit fast endophilin-mediated endocytosis

Antonio P. A. Ferreira [1,5,9], Alessandra Casamento [1,9], Sara Carrillo Roas [1], Els F. Halff [2,6], James Panambalana[1], Shaan Subramaniam[1,3], Kira Schützenhofer[1], Laura Chan Wah Hak[1,7], Kieran McGourty [1,8], Konstantinos Thalassinos[1], Josef T. Kittler [2], Denis Martinvalet [4] & Emmanuel Boucrot [1,3✉]

Endocytosis mediates the cellular uptake of micronutrients and cell surface proteins. Fast Endophilin-mediated endocytosis, FEME, is not constitutively active but triggered upon receptor activation. High levels of growth factors induce spontaneous FEME, which can be suppressed upon serum starvation. This suggested a role for protein kinases in this growth factor receptor-mediated regulation. Using chemical and genetic inhibition, we find that Cdk5 and GSK3β are negative regulators of FEME. They antagonize the binding of Endophilin to Dynamin-1 and to CRMP4, a Plexin A1 adaptor. This control is required for proper axon elongation, branching and growth cone formation in hippocampal neurons. The kinases also block the recruitment of Dynein onto FEME carriers by Bin1. As GSK3β binds to Endophilin, it imposes a local regulation of FEME. Thus, Cdk5 and GSK3β are key regulators of FEME, licensing cells for rapid uptake by the pathway only when their activity is low.

[1] Institute of Structural and Molecular Biology, University College London, London, UK. [2] Department of Neuroscience, Physiology, and Pharmacology, University College London, London, UK. [3] Institute of Structural and Molecular Biology, Birkbeck College, London, UK. [4] Department of Biochemistry, University of Padova, Padova, Italy. [5] Present address: Department of Pathology, Brigham and Women's Hospital, Harvard Medical School, Boston, MA, USA. [6] Present address: Department of Psychosis Studies, Institute of Psychiatry, Psychology & Neuroscience, King's College London, London, UK. [7] Present address: Centre for Neural Circuits and Behaviour, University of Oxford, Oxford, UK. [8] Present address: Department of Chemical Sciences, Bernal Institute, University of Limerick, Limerick, Ireland. [9] These authors contributed equally: Antonio P. A. Ferreira, Alessandra Casamento ✉email: e.boucrot@ucl.ac.uk

Clathrin-mediated endocytosis (CME) is the major uptake pathway in resting cells[1,2], but additional Clathrin-independent endocytic (CIE) routes, including fast Endophilin-mediated endocytosis (FEME), perform specific functions or internalize specific cargoes[3,4]. FEME is not constitutively active but is triggered upon the stimulation of selected cell surface receptors by their ligands[5]. These include G-protein coupled receptors (e.g., β1-adrenergic receptor, hereafter β1AR), receptor tyrosine kinases (e.g., epidermal growth factor receptor, EGFR) or cytokine receptors (e.g., Interleukin-2 receptor)[5]. In resting cells, FEME is primed by a cascade of molecular events starting with active, GTP-loaded, Cdc42 recruiting CIP4/FBP17 that engage the 5′-phosphatase SHIP2 and Lamellipodin (Lpd). The latter then concentrates Endophilin into clusters on discrete locations of the plasma membrane[6]. In absence of receptor activation, the clusters dissemble quickly (after 5–15 s) upon local recruitment of the Cdc42 GTPase-activating proteins RICH1, SH3BP1, or Oligophrenin[6]. New priming cycles start nearby, constantly preparing the plasma membrane for FEME. Upon activation, receptors are quickly sorted into pre-existing Endophilin clusters that then bud to form FEME carriers, which are Clathrin-negative, Endophilin-positive assemblies (EPAs) found in the cytosol. The entire process takes 4–10 s[6]. These FEME carriers travel rapidly to fuse with early endosomes and deliver their cargoes. While it is well established how FEME carriers are produced, we know very little about the regulation of the pathway. Grown in their respective full-serum media, different cell types display various levels of FEME. And the removal of growth factors (serum starvation) shuts down the pathway. Thus, we reasoned that such growth factor receptor-mediated regulation could be controlled by protein kinases. Here, we show that Cdk5 and GSK3β are negative regulators of FEME. Their chemical or genetic inhibition is sufficient to activate FEME promptly in resting cells, and upon addition of dobutamine, to boost the production of FEME carriers containing β1AR. The kinases inhibit at least three steps: endophilin binding and recruitment of CRMP4 and Dynamin-1 and Bin1 binding and recruitment of Dynein. This regulates cargo sorting, membrane scission, and FEME carrier transport onto microtubules. In hippocampal neurons, this control is required for proper growth cone formation, axon elongation, and branching.

## Results

**Different cell types display various levels of FEME activity.** Some cell types display spontaneous FEME when grown in their regular culture medium, while others do not. Normal RPE1 cells, primary *human* dermal fibroblasts (hDFA) and *human* umbilical vein endothelial cells (HUVEC) exhibited robust FEME in resting cultures grown in regular media (~5 to 15 EPAs per 100 μm$^2$, Suppl. Fig. 1a–d). In contrast, HeLa, HEK293, and BSC1 cells displayed very little spontaneous FEME. In these cells, not only was FEME identified in a minority of cells, but also a low number of FEME carriers were detected in those that were active (~1 to 3 EPAs per 100 μm$^2$, near the leading edge, Fig. S1a–c). This is not to be confused with FEME priming events (Endophilin short-lived clustering without subsequent carrier budding), which are identified by live-cell microscopy by the growing and disappearance of Endophilin spots (blinking) without any lateral movements. For example, BSC1 cells display abundant priming (clustering of Endophilin) but very little spontaneous FEME (fast moving EPAs into the cytosol)[5,6]. However, within a same culture, not all the cells displayed FEME (the maximum was ~60% of HUVEC cells, Fig. S1a, b, d).

In all cell types tested, FEME was enhanced by the addition of 10% serum to complete medium, as shown by an increase in the percentage of cells showing FEME activity as well as increase in EPA production (Suppl. Fig. 1b, c). Because FEME was inactive in cells starved of serum[5] (i.e., without growth factors), we looked for kinases that may regulate its activity. Other endocytic pathways are regulated both positively and negatively by multiple protein kinases[7–10]. Thus, we investigated whether phosphorylation would trigger or hinder FEME and looked for mechanisms that may control the level of FEME activity.

**Inhibition of Cdk5 and GSK3 activates FEME.** Small molecule inhibitors were used to screen the role of kinases known to regulate membrane trafficking and actin cytoskeleton dynamics (which is required for FEME). The compounds used were amongst the best-reported inhibitors for each kinase[11,12]. Four concentrations were tested (10, 100 nM and 1, 10 μM), with the minimum effective concentrations for each compound selected for further measurements. Small compounds were chosen because they can be used for short timeframes (minutes), limiting indirect effects on other kinases and long-term cumulative trafficking defects induced by gene depletion. The cells were treated for 10 min at 37 °C with the inhibitors diluted in regular growth medium containing 10% serum, and then fixed with pre-warmed paraformaldehyde solution (to preserve FEME carriers, see "Methods" section). RPE1 cells were used because they display robust spontaneous FEME when grown in regular medium (Fig. 1a, normal). This allowed identification of kinase inhibitors that either decreased or increased FEME. Normal FEME activity (that is, similar to DMSO-treated cells) was given one mark during scoring (Fig. 1a). Positive and negative controls (dobutamine and PI3Ki, respectively[5]) benchmarked the scoring for decreased (zero mark) and increased (two marks) FEME (Fig. 1a). Decreased FEME was assigned for samples with >80% reduction in the number of EPAs, in at least 50% of the cells. Increased FEME was attributed to samples with >200% elevation in the number of EPAs, in at least 50% of the cells.

The stringent criteria used likely missed mild modulations but revealed robust regulators of FEME. Inhibition of CaMKK1 and 2, SYK, FAK, or mTORC1/2 reduced FEME significantly (Fig. 1b), but this was not investigated further in the present study. Conversely, acute inhibition of Cdk5, GSK3 or p38 increased spontaneous FEME (Fig. 1b). Even though the three inhibitors for Cdk5 also inhibited other cyclin-dependent kinases, we were able to exclude a role for Cdk1 and 2 as inhibitors of these kinase failed to enhance FEME activity (Fig. 1b), and because Cdk7 and 9 are nuclear[13]. The role for p38 in FEME was not investigated further at this stage. Cdk5 inhibitors, Dinaciclib, and Roscovitine, and GSK3 inhibitors, BIO and CHIR-99021, were validated to enhance FEME in a dose-dependent manner (Fig. 1c), confirming that these kinases inhibit FEME in resting cells. Inhibition of Cdk5 and GSK3 increased productive FEME, as a larger number of endocytic carriers contained β1AR upon its activation by dobutamine (Fig. 1d). This is also coherent with the overlap of the binding sites of heterotrimeric G proteins and Endophilin on β1AR[14,15].

**Endophilin recruits GSK3β to regulate FEME locally.** GSK3α and β kinases require prior phosphorylation of their substrates by several other kinases (e.g., PKA, AMPK, CK1/2, and Cdk5). GSK3 docks onto these primed sites and then phosphorylates nearby Serines or Threonines, a few amino acids away[16]. The kinase is auto-inhibited by phosphorylation at its N-terminus (Ser 9 in GSK3β, pS9-GSK3β hereafter), which then occupies its docking site and thus blocks its interaction with substrates. This phosphorylation on GSK3 is mediated by many kinases that are activated by growth factor receptor signaling including AKT and

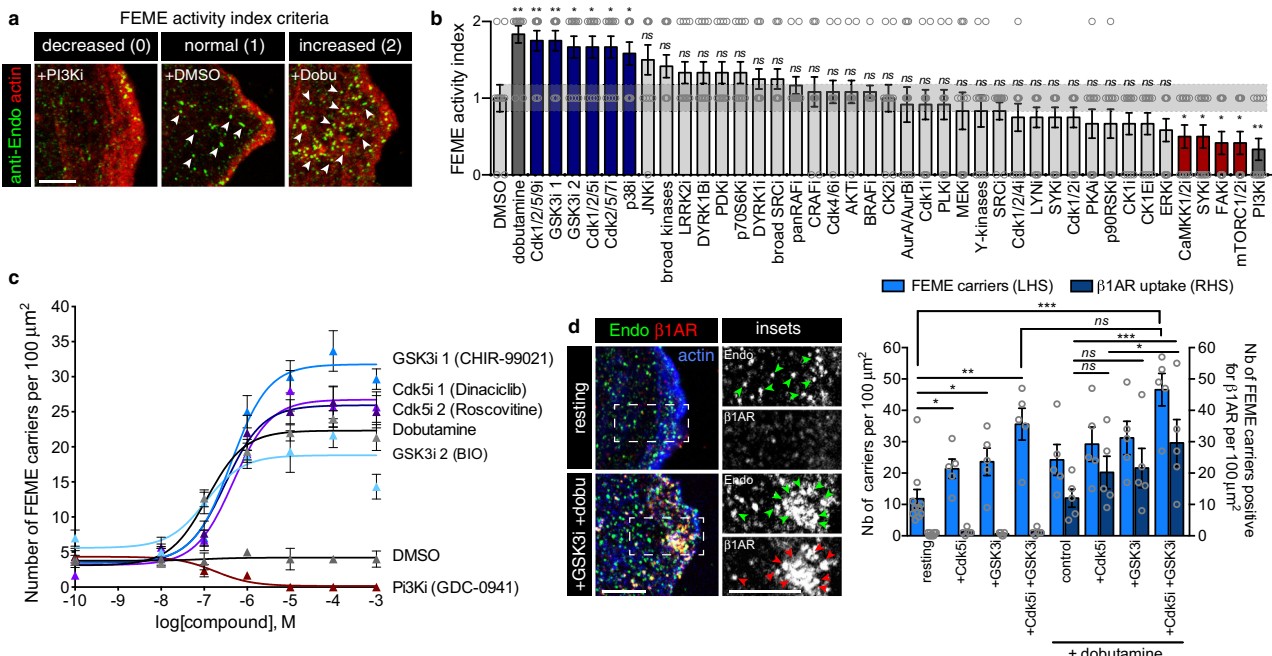

**Fig. 1 Acute inhibition of Cdk5 and GSK3 activates FEME. a** Scoring criteria used in the kinase screen. Representative images of decreased, normal, and increased FEME in resting human RPE1 cells treated with 10 μM dobutamine, 10 μM DMSO, and 10 nM GDC-0941 (PI3Ki), respectively. Arrowheads point at FEME carriers. Decreased FEME was assigned for samples with >80% reduction in the number of cytoplasmic Endophilin-positive assemblies (EPAs), in at least 50% of the cells. Increased FEME was attributed to samples with >200% elevation in the number of EPAs, in at least 50% of the cells. The corresponding scoring marks were 0, 1, and 2, respectively. Scale bar, 5 μm. **b** Kinase screen using small compound inhibitors. RPE1 cells grown in complete medium were incubated for 10 min at 37 °C with the following inhibitors: DMSO, (vehicle); dobutamine, 10 μM (positive control); Dinaciclib (Cdk1/2/5/9i), 1 μM; CHIR-99041 (GSK3i1), 1 μM; BIO (GSK3i2), 1μM; Roscovitine (Cdk1/2/5i), 1 μM; PHA-793887 (Cdk2/5/7i), 100 nM; VX-745 (p38i), 10 μM; JNK-IN-8 (JNKi), 1 μM; staurosporine (broad kinases), 1 μM; GNE-7915 (LRRK2i), 1 μM; AZ191 (DYRK1Bi), 10 μM; GSK2334470 (PDKi), 10 μM; PF-4708671 (p70S6Ki), 10μM; AZ191 (DYRKi), 10 μM; AZD0530 (broad SRCi), 1 μM; TAK-632 (panRAFi), 10 μM; GW 5074 (CRAFi), 1μM; PD0332991 (Cdk4/6i), 1 μM; MK2206 (AKTi), 1 μM; GDC-0879 (BRAFi), 1 μM; CX-4945 (CK2i), 1 μM; ZM 447439 (AurA/AurBi), 1 μM; RO-3306 (Cdk1i), 100 nM; BI 2536 (PLKi), 1 μM; PD0325901 (MEKi), 100 nM; Genistein (Y-kinases), 1 μM; Purvalanol A (Cdk1/2/4i), 100 nM; MLR 1023 (LYNi), 1 μM; P505-15 (SYKi), 100 nM; CDK1/2 inhibitor III (Cdk1/2i), 100 nM; KT 5720 (PKAi), 100 nM; BI-D1870 (p90RSKi), 100 nM; D4476 (CK1E), 1 μM; PF-4800567 (CK1Ei), 1 μM; SCH772984 (ERKi), 100 nM; STO609 (CaMKK1/2ii), 100 nM; P505-15 (SYKi), 1μM; PND-1186 (FAKi), 100 nM; Torin 1 (mTORC1/2i), 10 μM and GDC-0941 (PI3Ki), 100 nM (negative control). Histograms show the mean ± SEM from 12 well per condition, from three independent biological experiments. Statistical analysis was performed by one-way ANOVA. *ns* non significant; *$P < 0.05$, **$P < 0.01$. **c** Number of FEME carriers (EPAs) upon titration of CHIR-99021, BIO, Roscovitine and Dinaciclib. Dobutamine and GDC-0941 were used as positive and negative controls, respectively. Plots show the mean ± SEM from three cells per condition and per timepoint, from three independent biological experiments. **d** β1-adrenergic receptor (β1AR) uptake into FEME carriers in RPE1 cells pre-treated with 5 μM CHIR-99021 (GSK3i) for 5 min, followed by 10 μM dobutamine for 4 min or not (resting). Scale bars, 5 μm. Histograms show the mean ± SEM of the number of FEME carriers (LHS: left hand side) and the number of FEME carriers positive for β1AR per 100 μm² (RHS: right hand side) ($n = 30$ cells per condition, from biological triplicates). Arrowheads point at FEME carriers. Statistical analysis was performed by two-way ANOVA. *ns* non significant; *$P < 0.05$, **$P < 0.01$, ***$P < 0.001$.

ERK[17,18]. In absence of growth factors (such as upon serum starvation), pS9-GSK3β is reduced, relieving the auto-inhibition of the kinase[18]. Consistently, we observed reduced levels of inactive pS9-GSK3β and depression of FEME in cells starved for growth factors (Fig. 2a, b). In contrast, stimulating cells with an additional 10% serum (20% final) for 10 min inactivated the kinase (as deduced from the high levels of inactive pS9-GSK3β), and activated FEME beyond resting levels (Fig. 2a, b). There was a linear correlation ($r^2 = 0.99$) between the levels of inactive GSK3β and the numbers of FEME carriers within the same cells (Fig. 2a, b). However, activation of other receptors (e.g., β1AR with dobutamine) activated FEME in resting cells without measurable changes in GSK3β activity (Fig. 2b, +dobutamine, middle). There was a maximum level of FEME activity, as addition of dobutamine on cells that were previously activated with 10% extra serum did not increase the number of EPAs further (Fig. 2b, +dobutamine, right). When GSK3β activity was high (starved cells), dobutamine could not activate FEME (Fig. 2b, +dobutamine, left). There was a poor correlation ($r^2 = 0.67$) between the kinase activity and FEME

stimulation by dobutamine, suggesting that GSK3β acts upstream, imposing a cap on FEME that can be quickly lifted upon receptor activation.

As mass spectrometry detected GSK3β amongst the proteins immunopurified by anti-Endophilin antibodies (Suppl. Fig. 2a and Source Data file), we further characterized its binding to Endophilin. We found that GSK3β, but not α, bound to the SH3 domain of Endophilin but not that of the closely related N-BAR domain protein Bin1 (Fig. 2c). Consistently with the detection of GSK3β from resting but not FBS-stimulated extracts, (Suppl. Fig. 2a), the binding was not detected in extracts of cells that were inhibited for Cdk5 and GSK3 prior to lysis (Fig. 2c). Furthermore, GSK3β was detected on priming Endophilin spots at the leading edge but reduced on FEME carriers following inhibition (Fig. 2d). In contrast, inactive GSK3β was detected more on EPAs than on priming spots at the leading edge (Fig. 2d). Consistent with binding data, Cdk5 and GSK3 inhibitors blocked the recruitment of both total and inactive GSK3β (Fig. 2d). Thus, we concluded that Endophilin recruits GSK3β to inhibit FEME locally. Importantly,

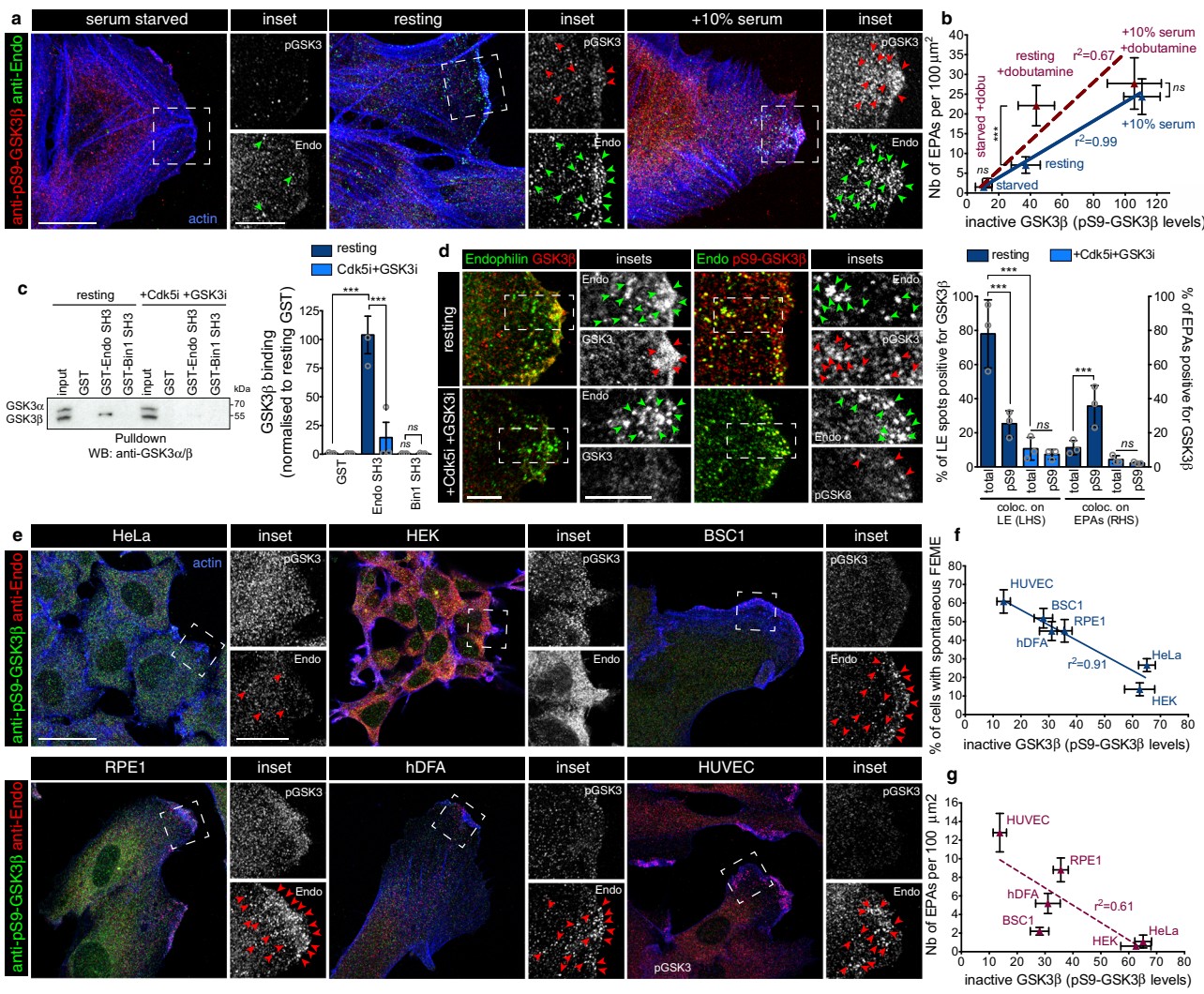

**Fig. 2 Endophilin recruits GSK3β for local regulation of FEME. a** Confocal images showing levels of phosphorylated Ser9 GSK3β (pS9-GSK3β) inactive kinase, and colocalization with Endophilin in cells starved of serum for 1 h (serum starved), grown in 10% serum medium (resting) or stimulated with additional serum for 10 min (+10% serum). Arrowheads point at cytoplasmic Endophilin-positive assemblies (EPAs), FEME carriers. Scale bar, 20 μm (main) and 10 μm (inset). **b** Correlation between the number of EPAs and pS9-GSK3β levels (single cell measurements) in cells that were starved of serum for 1 h (starved), grown in 10% serum medium (resting) or stimulated with additional serum for 10 min (+10% serum), followed by the addition of 10 μM dobutamine for 4 min (red data points) or not (blue data points). Plots show the mean ± SEM from five cells per condition from three independent biological experiments. Linear regression fit is indicated as $r^2$ values. Statistical analysis was performed by one-way ANOVA. *ns* non significant, ***$P < 0.001$.
**c** Pull-down experiments using beads with GST-SH3 domains of Endophilin A2 or Bin1, in resting cells or cells treated with 5 μM Dinaciclib (Cdk5i) and CHIR-99021 (GSK3i) for 10 min. GST beads were used as negative control. Bound GSK3β was detected using an antibody that detects both GSK3α and GSK3β. Histograms show the mean ± SEM of GSK3β binding, normalized to resting GST levels from three independent biological experiments. Statistical analysis was performed by one-way ANOVA. *ns* non significant, ***$P < 0.001$. **d** Colocalization of total and phosphorylated Ser9 GSK3β and Endophilin in cells treated with 5 μM Cdk5 and GSK3 inhibitors for 10 min, or not (resting). Arrowheads point at Endophilin spots and FEME carriers. Scale bars, 5 μm. Histograms show the mean ± SEM of Endophilin spots at the leading edge of cells (spots within 1 μm of cell edges, LHS: left hand side) and on EPAs (RHS: right hand side) positive for total or pS9-GSK3β. $n = 50$ spots or EPAs per condition, from three independent biological experiments. Statistical analysis was performed by one-way ANOVA. *ns* non significant, ***$P < 0.001$. **e** Confocal images showing levels of pS9-GSK3β and colocalization with Endophilin in resting HeLa, HEK, BSC1, RPE1, hDFA, or HUVEC grown in their respective full serum media. Arrowheads point at FEME carriers. Scale bar, 20 μm (main) and 10 μm (inset). **f** Correlation between the percentage of cells displaying active FEME and their pS9-GSK3β levels (single cell measurements) in the indicated resting cell types. Plots show the mean ± SEM from five cells per condition from three independent biological experiments. Linear regression fit is indicated as $r^2$ value. **g** Correlation between the number of EPAs and pS9-GSK3β levels (single cell measurements) in the indicated resting cell types. Plots show the mean ± SEM from five cells per condition from three independent biological experiments. Linear regression fit is indicated as $r^2$ value.

when comparing spontaneous FEME in different cell types, we found that levels of the inactive pS9-GSK3β were inversely correlated with the proportion of resting cells displaying spontaneous FEME ($r^2 = 0.91$, Fig. 2e, f). But there was no correlation between the kinase activity and the amounts of EPAs produced by spontaneous FEME in these resting cells ($r^2 = 0.61$, Fig. 2g).

**Cdk5 and GSK3β work in synergy to inhibit FEME.** Because dobutamine did activate FEME robustly, regardless of basal GSK3β activity, we hypothesized that there must be another layer of regulation upstream or parallel to the kinase. Given the requirement of a priming kinase for GSK3 action[16,19], we tested a potential synergy with Cdk5. Genetic inhibition of either Cdk5 or

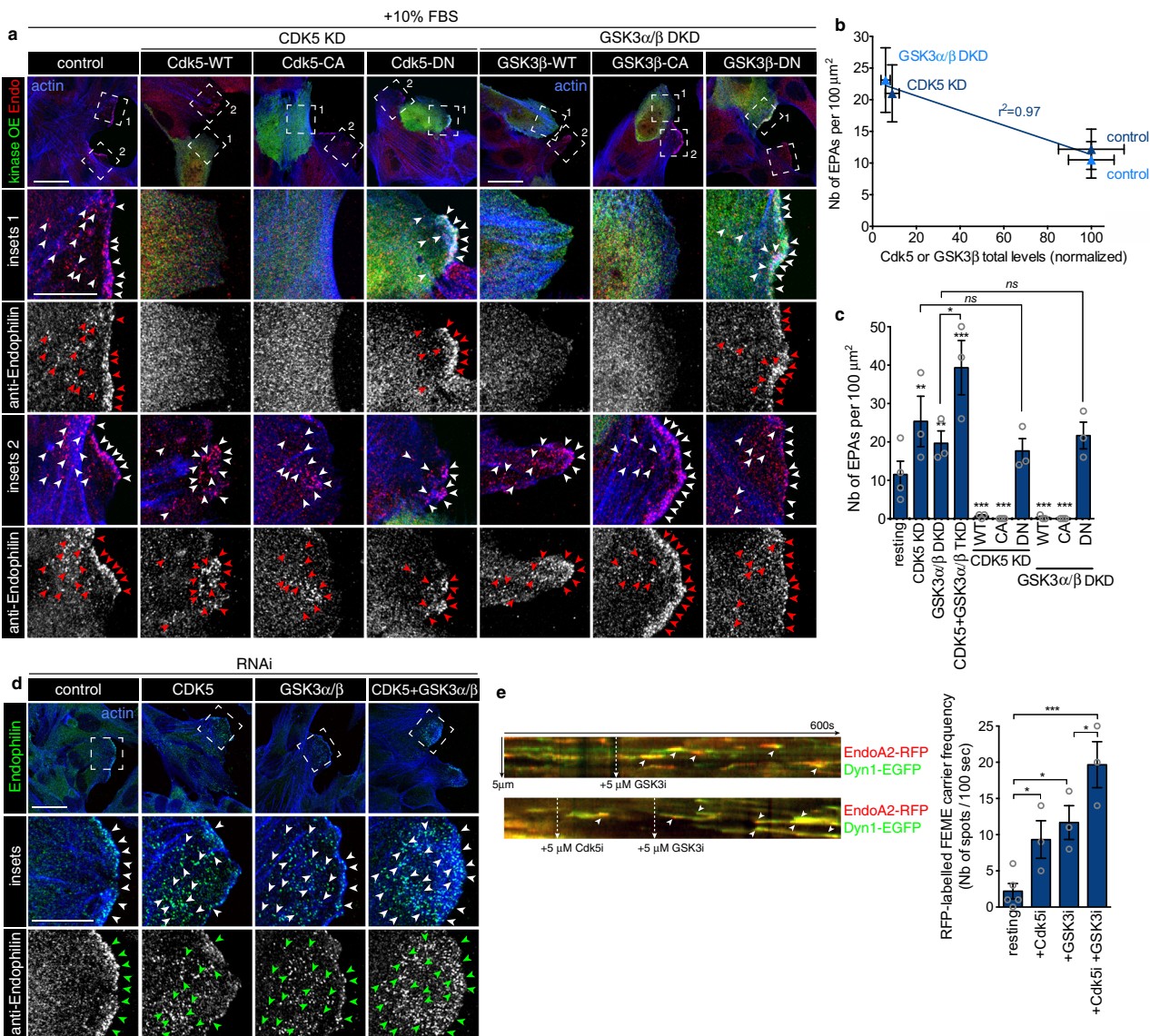

**Fig. 3 Cdk5 and GSK3β act in synergy to control FEME. a** Confocal microscopy images of control RPE1 cells (control), or cells in which Cdk5 or GSK3α and β had been knocked-down using RNAi (CDK5 KD or GSK3α/β DKD, respectively) cells. Wild-type (WT), constitutively active (CA) or dominant negative (DN) forms of Cdk5 or GSK3β (green) were overexpressed in a knock-down background and endogenous Endophilin (red) was immunostained, as indicated. All cells were stimulated with +10% FBS (20% final) for 10 min prior to fixation. Arrowheads point at FEME carriers. Scale bars, 20 μm. **b** Correlation between the number of EPAs and their Cdk5 or GSK3β levels (single cell measurements) in control, Cdk5 or GSK3α/β depleted cells, as indicated. Plots show the mean ± SEM from five cells per condition from three independent biological experiments. Linear regression fit is indicated as $r^2$ value. **c** Number of EPAs in RPE1 cells treated as indicated in **a** and **d**. Histograms show the mean ± SEM from three independent biological experiments ($n = 50$ per condition). Statistical analysis was performed by one-way ANOVA. *ns*, non significant; *$P < 0.05$, **$P < 0.01$, ***$P < 0.001$. **d** Confocal microscopy images of resting control RPE1 cells (control), Cdk5 (CDK5 KD), GSK3α and β (GSK3α/β DKD) or Cdk5 and GSK3α/β (Cdk5 + GSK3α/β TKD) knocked-down cells. Arrowheads point at FEME carriers. Scale bar, 20 μm (main) and 10 μm (inset). **e** Kymographs from cells expressing low levels of Dynamin1-EGFP and EndophilinA2-RFP, treated with CHIR-99021 (GSK3i) or Dinaciclib (Cdk5i) as indicated and imaged at 2 Hz. Arrowheads point at FEME carriers. Kymographs are representative of at least three captures from biological triplicates. Histograms show the mean ± SEM from three independent biological experiments ($n = 3$ cells per condition). Statistical analysis was performed by one-way ANOVA. *$P < 0.05$, ***$P < 0.001$.

GSK3α/β increased spontaneous FEME in resting cells (Fig. 3a, b and Suppl. Fig. 2b), and, conversely, in individual cells, the expression levels of the kinases correlated negatively with FEME activity ($r^2 = 0.97$, Fig. 3b). This confirmed the data obtained with the kinase inhibitors, but also suggests that as long as the kinases are inactive, FEME is elevated, as gene depletion by RNAi persists for several days. RNAi rescue with high levels of either wild type (WT) or constitutively active (CA) forms of the kinases were sufficient to suppress FEME. However, dominant-negative forms (DN) of either Cdk5 or GSK3β did not rescue the effect of the depletion of the endogenous kinases (Fig. 3a, c). The dual inhibition of Cdk5 and GSK3β occasioned a synergistic activation of FEME (Figs. 1d and 3d, e), driving not only cargo loading but also FEME carrier budding and lateral movement within the cytoplasm.

**Cdk5 and GSK3β regulate dynamin recruitment onto FEME carriers.** Dynamin mediates the budding of FEME carriers[5] and is known to be phosphorylated on Ser778 by Cdk5[20], and

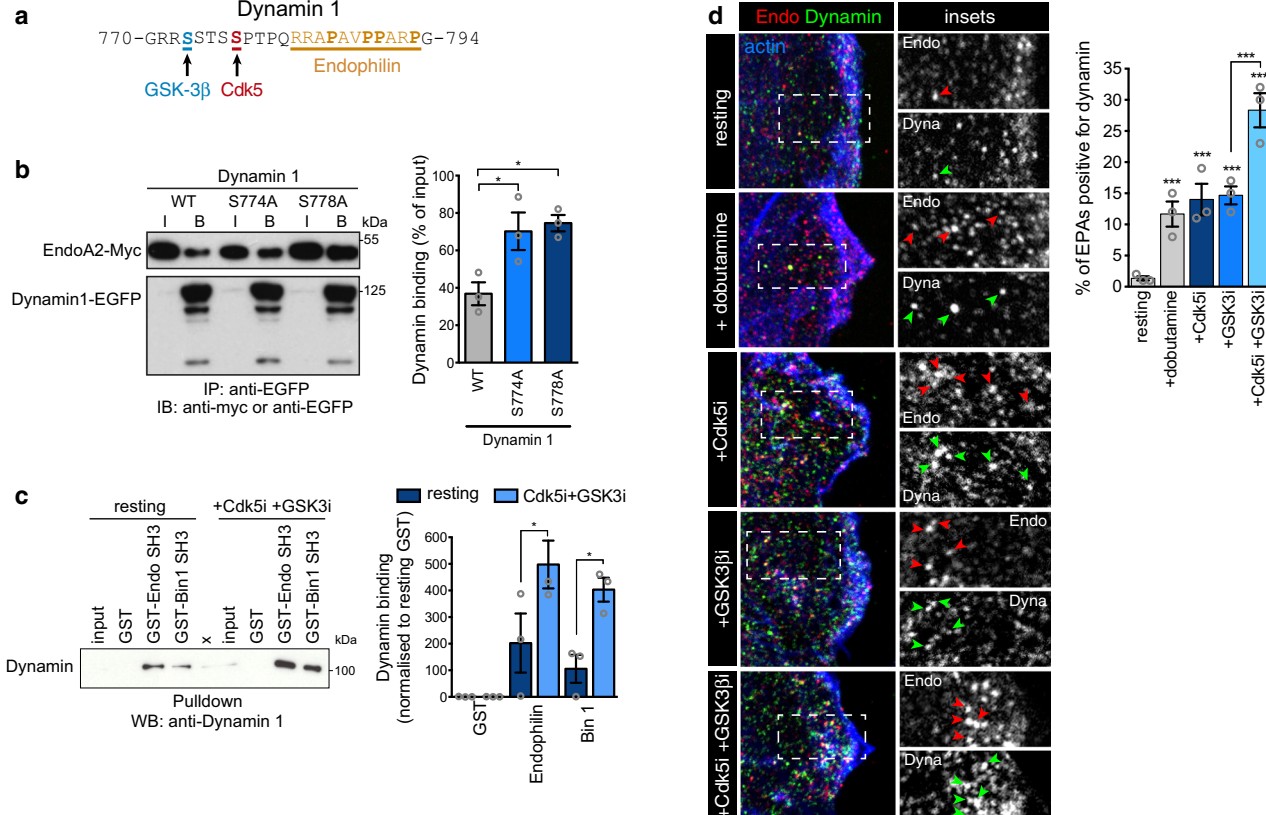

**Fig. 4 Cdk5 and GSK3β regulate Dynamin recruitment onto FEME carriers. a**, *Human* Dynamin-1 sequence (aa 770-794). Amino acids phosphorylated by GSK3 β (S774) and by Cdk5 (S778) are shown in blue and red, respectively. The proline-rich motif to which Endophilin is known to bind to is shown in orange (underlined). **b**, Co-immunoprecipitation of Endophilin A2-Myc and Dynamin1-EGFP wild-type (WT), or non-phosphorylatable mutants S774A or S778A. Inputs (I) correspond to 0.5% of the cell extracts), and bound fractions (B) to 90% of material immunoprecipitated. Right, Histograms show the mean ± SEM from three independent biological experiments. Statistical analysis was performed by one-way ANOVA. *$P < 0.05$. **c** Pull-down experiments using beads with GST-SH3 domains of Endophilin A2 or Bin1, in resting cells or cells treated with 5 μM Dinaciclib (Cdk5i) and CHIR-99021 (GSK3i) for 10 min. GST beads were used as negative control. X labels a lane that was not used in this study. Inputs correspond to 4% of cell extracts. Right, histograms show the mean ± SEM of Dynamin-1 binding, normalized to resting GST levels from three independent biological experiments. Statistical analysis was performed by two-way ANOVA. *$P < 0.05$. **d** Recruitment of endogenous Dynamin onto FEME carriers in RPE1 cells treated for 10 min with 5 μM Dinaciclib (Cdk5i) and/or CHIR-99021 (GSK3i) or not, followed by 10 μM dobutamine for 4 min. Arrowheads point at FEME carriers. Scale bars, 5 μm. Histograms show the mean ± SEM from three independent biological experiments ($n = 50$ cells per condition). Statistical analysis was performed by one-way ANOVA. ***$P < 0.001$.

subsequently on Ser774 by GSK3β (Fig. 4a). This blocks the function of Dynamin in CME but is required for activity-dependent bulk endocytosis in synapses[8,9]. Consistent with proximity of the phosphorylated residues to the binding site of Endophilin on Dynamin[21,22], the inhibition of both Cdk5 and GSK3 increased Dynamin recruitment onto budding FEME carriers (Figs. 3e and 4c, d). Furthermore, mutation of the Cdk5 and GSK3β phosphorylation sites on Dynamin also increased its binding to Endophilin (Fig. 4b). As the single inhibition of either Cdk5 or GSK3β was sufficient to relieve FEME from their control (even though the other kinase should not be affected), we concluded that Cdk5 acts upstream of GSK3β, and that other kinases may prime GSK3β in absence of Cdk5. The synergy of their inhibition suggested that they hamper FEME at several steps, beyond Dynamin recruitment.

**Cdk5 blocks the binding of Endophilin to CRMP4 and the sorting of PlexinA1 into FEME carriers.** Amongst proteins known to be phosphorylated by Cdk5 and GSK3β is collapsin response mediator protein 4 (CRMP4)[23]. We recently identified CRMP1, 3, and 4 in pulldown experiments using Endophilin SH3

domains[6]. CRMP1 to 5 form homo-tetramers and hetero-tetramers acting as adaptors during cell guidance mediated by Plexin A1 (Fig. 5a), and mediate cytoskeletal remodeling upon Semaphorin 3A or 6D sensing[24–26]. Cdk5 phosphorylates CRMP4 at Ser 522, which primes GSK3β-mediated phosphorylation at the positions Thr 509, Thr 514, and Ser 518[23] (Fig. 5a). The phosphorylation of CRMP4 perturbs its binding to microtubules and actin and CRMP4 is critical for proper neuronal development in both zebrafish and mice[27,28]. Inhibiting Cdk5, or overexpressing the non-phosphorylatable mutant CRMP4-S522A, increased CRMP4 binding to Endophilin, whereas a phospho-mimetic mutation S522D had the opposite effect (Fig. 5b, c, e, f and Suppl. Fig. 3a, b, d). Endophilin recruits cargoes into FEME carriers through the binding of its SH3 domain to proline rich motifs present in cargo adaptors or cytoplasmic tails of receptors[5]. Point mutations P526A or R525E, but not P502A, abolished the interaction with Endophilin (Fig. 5d and Suppl. Fig. 3c), thus establishing that the binding motif for Endophilin on CRMP4 is the proline-rich motif (aa 523-529) proximal to Ser 522 (Fig. 5a). The sites phosphorylated by GSK3β are several amino acids away from that motif, explaining why GSK3 inhibition did not increase the interaction of Endophilin with CRMP4

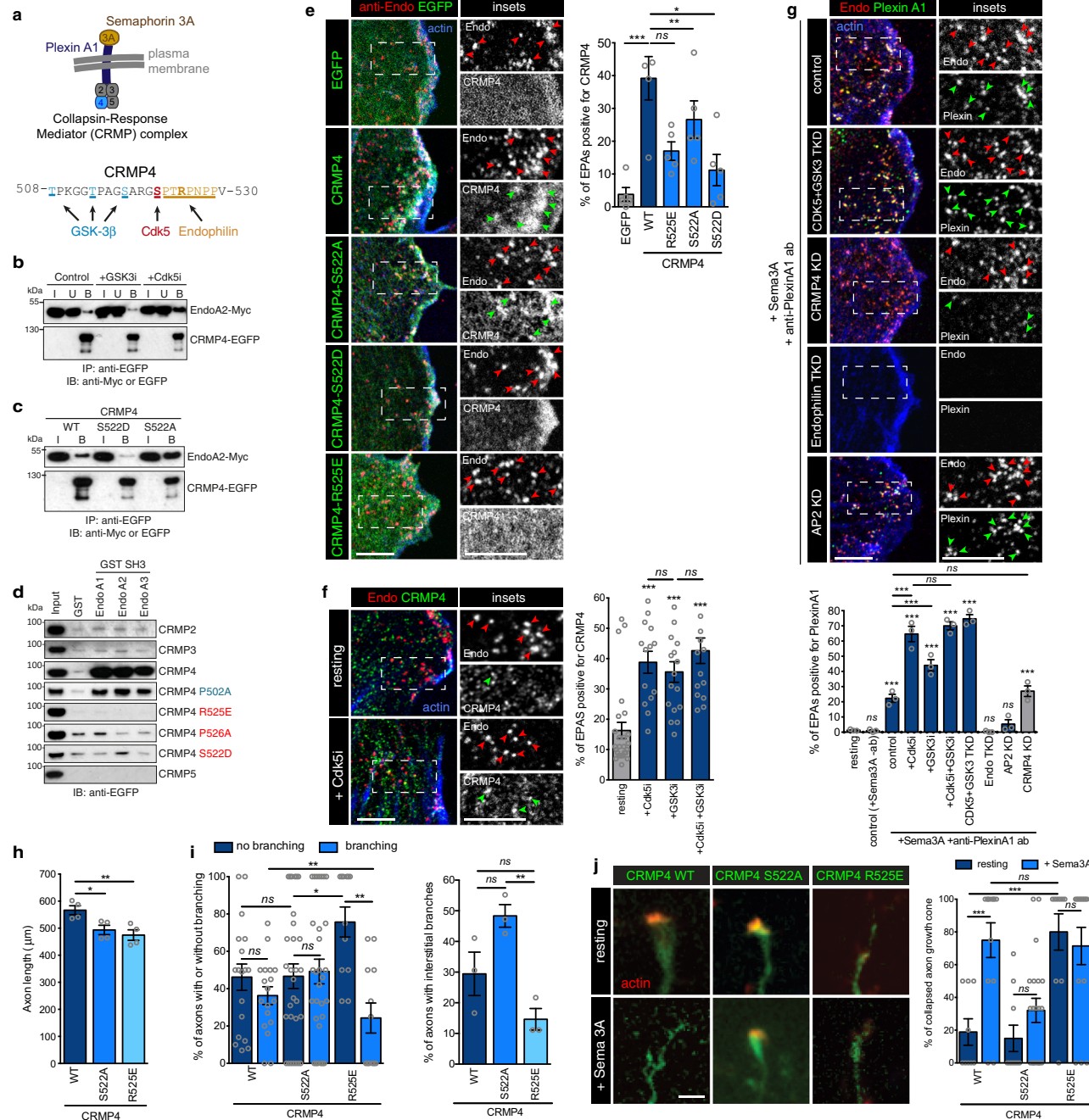

(Fig. 5b and Suppl. Fig. 3a). However, inhibiting GSK3 increased their colocalization in cells (Fig. 5f and Suppl. Fig. 3e), probably through the broad activation of FEME. Consistent with the biochemical data, mutations in CRMP4 inhibiting its binding to Endophilin abrogated localization of CRMP4 on FEME carriers, even upon co-overexpression (Fig. 5e and Suppl. Fig. 3d).

In HUVEC cells, which express CRMP4 and Plexin A1 endogenously[29], both Cdk5 and GSK3 inhibition enhanced the recruitment of endogenous CRMP4 onto FEME carriers and the uptake of Plexin A1 upon Semaphorin 3A stimulation (Fig. 5f, g and Suppl. Fig. 3e–g). However, the dual inhibition of Cdk5 and GSK3 did not induce a synergistic effect on Endophilin-CRMP4 colocalization nor PlexinA1 uptake (Fig. 5f, g and Suppl. Fig. 3e, g). Similarly to other receptors (e.g., EGFR)[5], PlexinA1 can be internalized both via FEME and CME to enter cells (Suppl. Fig. 3f), perhaps from different cellular location. At the leading edge of

HUVEC cells, where FEME is prominent (Suppl. Fig. 1d), uptake of PlexinA1 upon Cdk5 inhibition was primarily through FEME carriers (Fig. 5g and Suppl. Fig. 3g).

Plexin A1 senses repulsive axon guidance cues such as Semaphorin 3A, which repels axons and inhibits axon elongation and branching[30–32]. Upon binding to secreted Semaphorin 3A, Plexin triggers local actin and microtubule depolymerization, thereby causing axon growth cone (AGC) collapse and abortion of axon branching[32]. As FEME cargoes accumulate at the plasma membrane upon inhibition of the pathway[5], we reasoned that unlinking Plexin A1 from FEME upon overexpression of the CRMP4 mutant that cannot bind to Endophilin would sensitize neurons to the repelling action of Semaphorin 3A.

At AGCs of *mouse* hippocampal neurons, we found that endogenous Plexin A1 located within FEME carriers in a proportion similar to that in resting HUVEC cells (~30%, Suppl.

**Fig. 5 Cdk5-mediated phosphorylation of CRMP4 inhibits the binding of Endophilin to CRMP4 and the sorting of Plexin A1 into FEME carriers. a** Top, diagram showing the recruitment of CRMP2-5 adaptor complex (CRMP4 highlighted in blue) to Plexin A1 upon stimulation with Semaphorin 3A. Bottom, *human* CRMP4 protein sequence (aa 508–530). Amino acids phosphorylated by GSK3β (T509, T514, and S518) and by Cdk5 (S522) are shown in blue and red, respectively. The Endophilin binding motif established in this study is shown in orange (underlined). **b** Co-immunoprecipitation of CRMP4-EGFP and Endophilin A2-Myc from cells treated with 5 μM CHIR-99021 (GSK3i) or Dinaciclib (Cdk5i) as indicated. I, input (10% of the cell extracts), U, unbound (10% of total) and B, bound fractions (80% of total), respectively. **c** Co-immunoprecipitation of CRMP4-EGFP wild-type (WT), S22D or S522A and Endophilin A2-Myc. I, input (10% of the cell extracts), and B, bound fractions (90% of total), respectively. **d** Pull-down using GST-SH3 domains of Endophilin A1, A2, or A3 and cell extracts expressing the indicated EGFP-tagged CRMP proteins. GST was used as negative control. Binding proteins were detected by immunoblotting with an anti-EGFP antibody. input lanes correspond to 5% of the cell extracts. **e** Recruitment of EGFP-tagged CRMP4 WT, S522A, S522D, or R525E onto FEME carriers (cytoplasmic Endophilin-positive assemblies, EPAs) in HUVEC cells. Arrowheads point at FEME carriers. Scale bars, 5 μm. Histograms show the mean ± SEM from three independent biological experiments (*n* = 30 cells per condition). Statistical analysis was performed by one-way ANOVA; ns non significant; *P < 0.05, **P < 0.01, ***P < 0.001. **f** Recruitment of endogenous CRMP4 onto FEME carriers in HUVEC cells treated for 10 min with 5μM Dinaciclib (Cdk5i) and/or CHIR-99021 (GSK3i), or left untreated (resting). Arrowheads point at FEME carriers. Scale bars, 5 μm. Histograms show the mean ± SEM from three independent biological experiments (*n* = 45 cells per condition). Statistical analysis was performed by one-way ANOVA; ns non significant; ***, P < 0.001. **g** Endogenous Plexin A1 uptake into FEME carriers in HUVEC cells depleted of Endophilin A1, A2, and A3 (Endophilin TKD), Cdk5 and GSK3α and β (CDK5 + GSK3α/β TKD), CRMP4 (CRMP4 KD) or AP2 (AP2 KD) or pre-treated with Cdk5i and/or GSK3i for 5 min. Cells were stimulated by 20 nM Semaphorin 3A (Sema3A) for 5 min in presence of 10 μg/mL anti-PlexinA1 antibodies (recognizing the ectodomain of PlexinA1) or left untreated (resting). Arrowheads point at FEME carriers. Scale bars, 5 μm. Histograms show the mean ± SEM from three independent biological experiments (*n* = 30 cells per condition). Statistical analysis was performed by one-way ANOVA; ns non significant; ***P < 0.001. **h** Axon length of *mouse* hippocampal neurons expressing EGFP-CRMP4 wild type (WT), EGFP-CRMP4 S522A or EGFP-CRMP4 R525E mutants. Histograms show the mean ± SEM of 73, 93, and 38 axons from 4 independent biological experiments, respectively. Statistical analysis was performed by one-way ANOVA; *, P < 0.05, **, P < 0.01. **i**, Total (Left) or Interstitial (Right) axon branching of *mouse* hippocampal neurons expressing EGFP-CRMP4 wild type (WT), EGFP-CRMP4 S522A or EGFP-CRMP4 R525E mutants. Histograms show the mean ± SEM of 12, 12 and 24 captures from 3 independent biological experiments, respectively. Statistical analysis was performed by one-way (Right) two-way (Left) ANOVA; ns non significant; *P < 0.05, **P < 0.01. **j** Axon growth cone collapse in resting or neurons treated with 5 nM of semaphorin 3A and expressing EGFP-CRMP4 wild type (WT), EGFP-CRMP4 S522A or EGFP-CRMP4 R525E mutants. Scale bar, 20 μm. Histograms show the mean ± SEM of 13, 10, and 12 captures from three independent biological experiments, respectively. Statistical analysis was performed by two-way ANOVA; ns non significant; ***P < 0.001.

Fig. 3h). Overexpression of the CRMP4 mutant that cannot bind to Endophilin (R525E) reduced axon length in *mouse* hippocampal neurons in comparison to WT CRMP4 (Fig. 5h and Suppl. Fig. 3i). Unlinking Plexin A1 from FEME upon CRMP4-R525E overexpression also caused a reduction in interstitial axon branching (also called collateral branching) as well as constitutive collapse of axon growth cones, even in absence of additional Semaphorin 3A (Fig. 5i, j and Suppl. Fig. 3i). The increased AGC collapse and strong axon growth and branching defects may be due to increased levels of Plexin A1 remaining at the cell surface, as its internalization is hampered by the absence of FEME carriers. At constant Semaphorin 3A concentrations, this would render neurons more sensitive to its repulsive activity.

Conversely, expression of the CRMP4 mutant that cannot be phosphorylated by Cdk5 (S522A) had no effect on axon branching. but protected AGCs from Semaphorin 3A-induced collapse (Fig. 5i, j and Suppl. Fig. 3i).

The CRMP4-S522A mutant likely caused a similar increase in Plexin A1 uptake compared to the one measured in HUVEC (Fig. 5g). By having less Plexin A1 on their surface (due to increased FEME), neurons expressing the S522A mutant would become less sensitive to the repelling action of Semaphorin 3A. Altogether, these experiments established that Cdk5 controls the binding of Endophilin to CRMP4, thereby regulating the uptake of Plexin A1 into FEME carriers and proper axon branching and extension.

To confirm these results with other axon guidance receptors, we tested interaction of receptors comprising putative proline-rich motifs in their cytoplasmic tails. We found that Endophilin binds to Semaphorin 6A, Semaphorin 6D, and ROBO1 cytoplasmic tails (Suppl. Fig. 4a). Roundabout (ROBO) receptors bind to Slit ligands to mediate cell guidance, including axon repulsion[33]. Recently, Endophilin was found to mediate the uptake of ROBO1 and VEGFR2 via a Clathrin-independent pathway reminiscent to FEME[34]. We confirmed that Slit1 enters cells via FEME carriers (Suppl. Fig. 4b) and its cellular uptake was

strongly reduced in FEME-deficient but not in CME-deficient cells (Endophilin triple knock-down, TKD, and AP2 knock-down, respectively, Suppl. Fig. 4c). Upon binding to Slit, ROBO1 activates AKT, which in turn phosphorylates GSK3β on Ser9, thereby de-activating the kinase in axons[35,36]. Consistently, acute inhibition of GSK3β increased the uptake of Slit1 into FEME carriers two-fold (Suppl. Fig. 4c). Thus, Cdk5 and GSK3β kinases act at another level of FEME by controlling the sorting of cargoes, such as PlexinA1 and ROBO1, into endocytic carriers.

**Cdk5 and GSK3β control the recruitment of Dynein by Bin1 onto FEME carriers.** As Cdk5 and GSK3 kinases regulate Dynein activity and binding to cargoes[37,38], and because FEME carriers containing Shiga toxin rely on this microtubule motor for scission and retrograde trafficking[39,40], a role for the kinases in regulating Dynein during FEME was explored. We confirmed that inhibiting Dynein (with either the small inhibitor Ciliobrevin D[41] or the overexpression of the p50 dynamitin subunit of the dynactin complex[42]) blocked FEME carrier budding and β1AR endocytosis (Suppl. Fig. 5a, b). However, inhibition of Kinesin upon over-expression of the TPR subunit[43] had no effect (Suppl. Fig. 5a, b) This is consistent with the role of microtubule motors in generating membrane tension, thus facilitating the scission of budding membrane carriers by Dynamin and BAR domain proteins such as Endophilin[44]. Most FEME carriers were detected in the vicinity of microtubules, and their mild depolymerization using low doses (100 nM) of nocodazole stalled FEME carriers at the plasma membrane (Fig. 6a). FEME carriers, produced in resting cells upon acute Cdk5 and GSK3β inhibition, recruited Dynein and travelled along microtubules (Fig. 6a–d). Dynein was immunopurified together with Endophilin from cell extracts in which Cdk5 was inhibited (Fig. 6e). To confirm that Dynein was recruited onto FEME carriers produced upon Cdk5 and GSK3β inhibition, immunoprecipitation was performed on membrane fractions enriched in Endophilin but containing only small amounts of other endocytic markers (Fractions 7 of sucrose

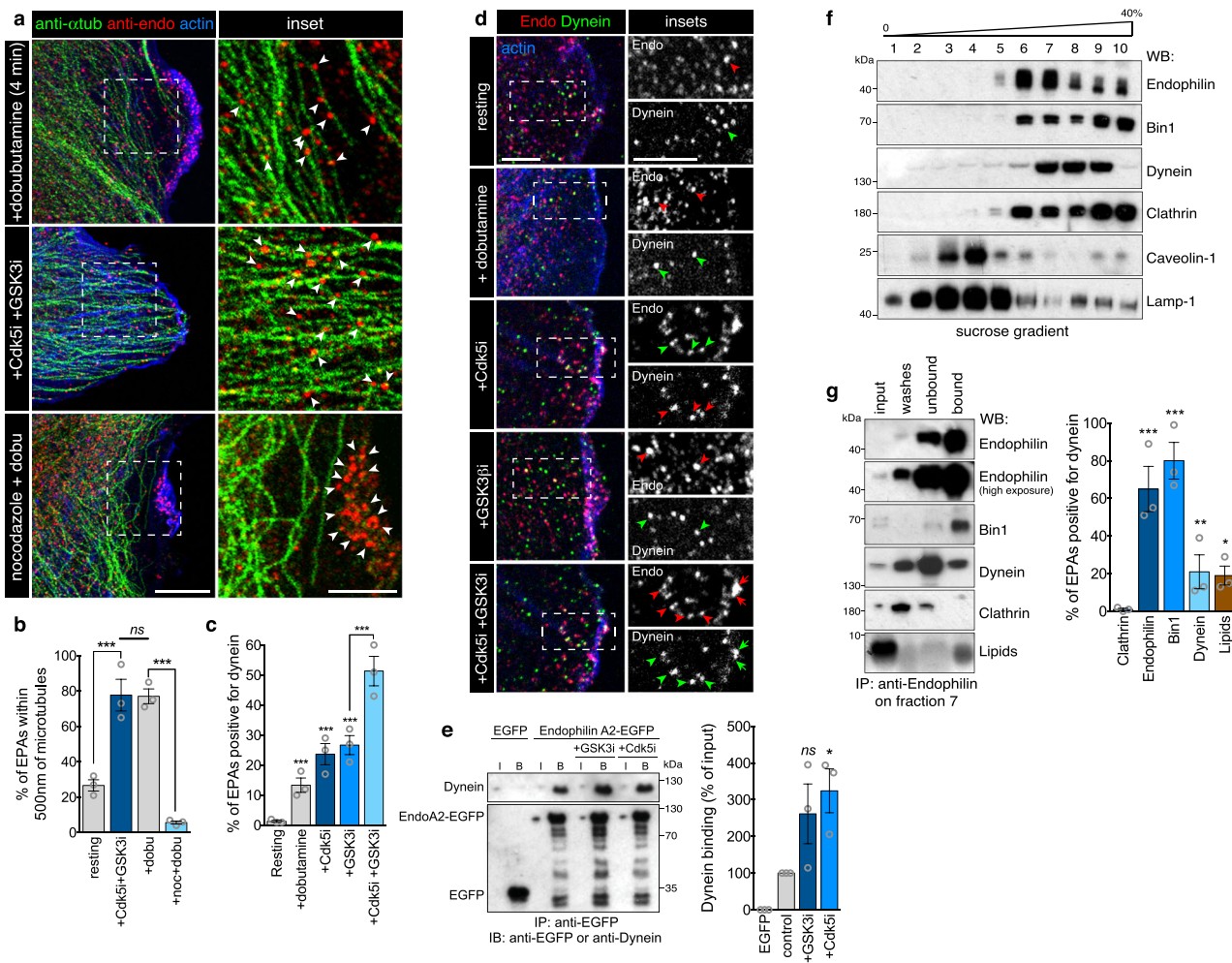

**Fig. 6 Cdk5 and GSK3β inhibit Dynein recruitment onto FEME carriers. a**, **b** Juxtaposition of FEME carriers and microtubules in HUVEC cells treated with 10 μM dobutamine for 4 min, 5μM Dinaciclib (Cdk5i) and CHIR-99021 (GSK3i) for 10 min, but not upon mild depolymerization (using 100 nM nocodazole for 10 min prior to dobutamine stimulation). Arrowheads point at FEME carriers. Scale bar, 20 μm (main) and 10 μm (inset). Histograms show the mean ± SEM from three independent biological experiments (*n* = 100 puncta per condition). Statistical analysis was performed by one-way ANOVA; *ns* non significant; ***P < 0.001. **c**, **d** Recruitment of endogenous Dynein onto FEME carriers in HUVEC cells treated for 10 min with 5 μM Dinaciclib (Cdk5i) and CHIR-99021 (GSK3i) or not, followed by 10 μM dobutamine for 4 min. Arrowheads point at FEME carriers. Scale bars, 5 μm. Histograms show the mean ± SEM from three independent biological experiments (*n* = 50 cells per condition). Statistical analysis was performed by one-way ANOVA; ***P < 0.001. **e** Co-immunoprecipitation experiments with EGFP or Endophilin A2-EGFP, in resting cells or cells treated for 30 min with 5μM Dinaciclib (Cdk5i) and CHIR-99021 (GSK3i). I, input (10% of the cell extracts), and B, bound fractions (90% of total), respectively. Histograms show the mean ± SEM from three independent biological experiments. Statistical analysis was performed by one-way ANOVA; *ns* non significant; *P < 0.05. **f** Sucrose gradient (0–40%) membrane isolation form RPE1 cells stimulated with 10% FBS (20% final) for 10 min. Fractions were immunoblotted for Endophilin, Bin1, Dynein, Clathrin, Caveolin-1, and Lamp-1. Fraction 7, containing high levels of Endophilin but low levels of Clathrin, Caveolin-1, and Lamp-1 was selected for subsequent immuno-precipitation. **g** Anti-Endophilin immuno-precipitation from fraction seven samples. Immunoblots measured the levels of Endophilin, Bin1, Dynein, Clathrin and lipids (detected using alcohol-free Coomassie[45], see "Methods" section) in input (5% of cell extracts), washes (10% of total), unbound (10% of total), and bound (50% of total) samples. Histograms show the mean ± SEM from three independent biological experiments.). Statistical analysis was performed by one-way ANOVA; *P < 0.05, **P < 0.01, ***P < 0.001.

gradients, Fig. 6f). The material that was immuno-isolated from such fractions likely contained FEME carriers, as they were rich in Endophilin but also in lipids (detected by alcohol-free Coomassie[45], see "Methods" section) but devoid of Clathrin. Importantly, Dynein was indeed immunopurified together with Endophilin from such fractions (Fig. 6f, g).

We were intrigued by the presence of Bin1 in the immuno-precipitated fractions, as we initially included it as a control (Bin1 is a N-BAR and SH3 domain-containing protein related to Endophilin). To test for a potential role for Bin1 and other BAR proteins in FEME, we screened our library containing 72 full-

length *human* BAR proteins tagged with EGFP[6]. But instead of looking for BAR proteins colocalizing onto the transient Endophilin clusters at the leading edge[6], we focused on those that localized onto FEME carriers produced upon FBS addition (Fig. 7a). While ten BAR domain proteins (FAM92B, SH3BP1, ASAP1, SNX9, SNX33, CIP4, Pacsin2, PSTPIP1, and Nostrin) were significantly detected onto a subset (~15 to 45%) of EPAs, only Amphiphysin, Bin1 and Bin2 located to the majority (>50%) of FEME carriers (Fig. 7a, b). This partial localization of the ten aforementioned BAR proteins could indicate that they play a role only in discrete steps of FEME and/or interact with sub-

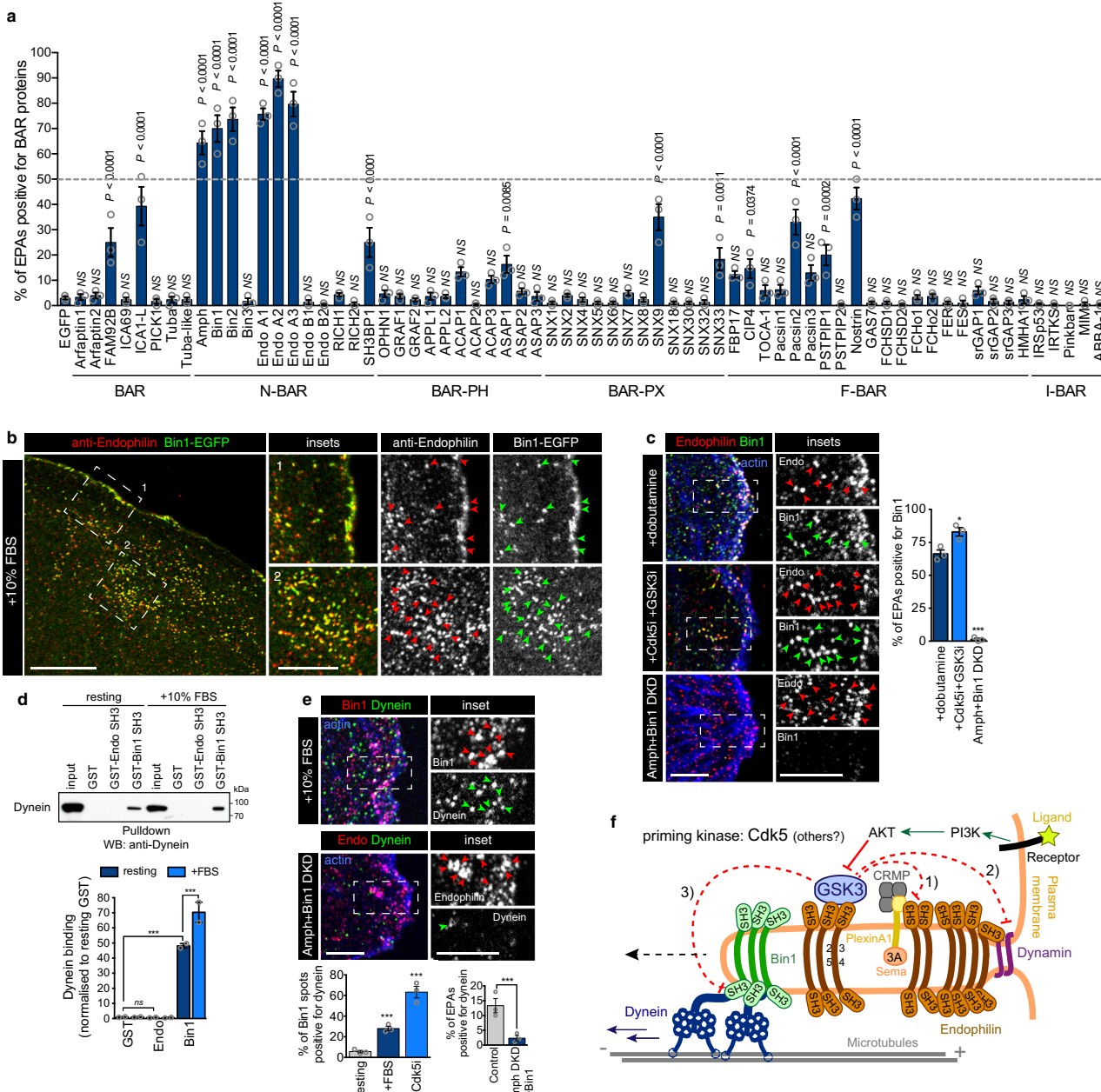

**Fig. 7 Bin1 recruits Dynein onto FEME carriers. a** Colocalization of named EGFP-tagged BAR proteins on FEME carriers marked by endogenous Endophilin in BSC1 cells stimulated with additional 10% serum for 10 min prior to fixation. Histograms show the mean ± SEM from three independent biological experiments (n > 100 puncta per condition). Statistical analysis was performed by one-way ANOVA; *NS* non significant; *P* values as indicated. **b** Colocalization of Bin1-EGFP on FEME carriers marked by endogenous Endophilin in BSC1 cells stimulated with additional 10% serum for 10 min prior to fixation. Arrowheads point at FEME carriers. Scale bar, 20 μm (main) and 10 μm (inset). **c** Colocalization of endogenous Bin1 and Endophilin upon stimulation with dobutamine, after treatment with 5 μM Dinaciclib (Cdk5i) and CHIR-99021 (GSK3i) for 10 min, or in cells depleted of Bin1 Amphiphysin (Amph + Bin1 DKD). Arrowheads point at FEME carriers. Scale bars, 5 μm. Histograms show the mean ± SEM from three independent biological experiments (n > 150 puncta per condition). Statistical analysis was performed by one-way ANOVA; *P < 0.05, ***P < 0.001. **d** Pull-down experiments using beads with GST-SH3 domains of Endophilin A2 or Bin1, in resting cells or cells treated with extra 10% FBS for 10 min. GST beads were used as negative control. Inputs correspond to 4% of cell extracts. Bottom, histograms show the mean ± SEM of Dynein binding, normalized to resting GST levels from three independent biological experiments. Statistical analysis was performed by two-way ANOVA; *ns* non significant; ***P < 0.001. **e** Colocalisation between Bin1 and Dynein in cells stimulated with extra 10% serum (top), and between endophilin and dynein upon Amphiphysin/Bin1 double knock-down (DKD)(bottom). Arrowheads point at FEME carriers. Scale bars, 5 μm. Histograms show the mean ± SEM from three independent biological experiments (n > 50 puncta per condition). Statistical analysis was performed by one-way ANOVA; ***, *P < 0.001*. **f** Model: multi-layered regulation of FEME by Cdk5 and GSK3β: 1) obstruction of CRMP4 binding to Endophilin and thus PlexinA1 sorting into FEME carriers upon Semaphorin 3A stimulation, 2) inhibiton of Dynamin recruitment onto FEME carriers, thus inhibiting vesicle budding and 3) hinderance of Dynein recruitment by Bin1, thereby reducing FEME carriers movement. GSK3β binds to Endophilin and acts locally to hold off FEME. In cells exposed to growth factors, PI3K-mediated signaling activates AKT and other kinases that controls GSK3β activity, and thus license cells for FEME.

populations of FEME carriers, or could be simply caused by the ectopic expression of the constructs. These proteins were not studied further at this point. Amongst the three best hits, we focused on Bin1 because it is ubiquitously expressed, unlike Amphiphysin, which is brain-enriched[46,47]. Bin2 is a known binding partner of Endophilin that is mainly expressed in leukocytes and that heterodimerizes with Bin1 but not Amphiphysin[48,49]. Bin1 has several splice variants, including brain-specific long isoforms 1–7 (also known as Amphiphysin-II) that contain AP2-binding and Clathrin-binding motifs and function in CME[50]. The two ubiquitously-expressed, short isoforms 9 and 10, however, resemble Endophilin in that they have a N-BAR domain, a short linker and SH3 domain and colocalize poorly with either AP2 or Clathrin (immunostaining, Suppl. Fig. 6a). Endogenous Bin1 localized onto the majority of FEME carriers produced upon either β1AR activation or Cdk5 and GSK3β inhibition (Fig. 7c). Like Endophilin[6], Bin1 binds to Lpd (Suppl. Fig. 6d) and relies on both CIP4 and Lpd for its recruitment into the transient clusters priming the leading edges of resting cells (Suppl. Fig. 6b, c). In cells depleted for Bin1 (Amphiphysin and Bin1 double knock-down, Amph+Bin1 DKD, was performed to avoid potential compensation), CIP4, Lpd, and Endophilin were recruited as in control cells (Suppl. Fig. 6b, c) and β1AR uptake was not affected[6]. This suggested that Bin1 could be mediating the uptake of different cargoes other than β1AR and/or that it could have a later role, postbudding. In absence of data supporting or rebutting the first hypothesis, we focused on the potential link with Dynein that we reported above.

Pull-down experiments revealed that the SH3 domain of Bin1 and not that of Endophilin isolated endogenous Dynein (Fig. 7d). As it was unlikely for Bin1 to bind directly to the motor domain, we tested Dynactin subunits that were identified by mass spectrometry in previous pull-downs[6]. However, neither Bin1 nor Endophilin bound to the p150glued or p27 subunits (Suppl. Fig. 6e). Stimulation of FEME by serum addition increased Dynein binding to Bin1 and recruitment onto FEME carriers (Fig. 7d, e and Suppl. Fig. 6f). Acute inhibition of Cdk5 and GSK3β increased the recruitment further, confirming that these kinases negatively regulate Dynein loading onto FEME carriers (Figs. 6c, d and 7e and Suppl. Fig. 6f). Remaining EPAs produced in cells depleted for Bin1 showed reduced colocalization with of Dynein (Fig. 7e), and often clustered at the cell periphery. Collectively, we showed that Bin1 recruits Dynein onto FEME carriers, under the control of Cdk5 and GSK3β. All our data taken together established these kinases as master regulators of FEME, antagonizing the process at several levels including cargo sorting, Dynamin and Dynein recruitment (Fig. 7f).

## Discussion

Cdk5 and GSK3β play important roles in regulating endocytosis. The best understood mechanism involves the phosphorylation of Dynamin-1 at Ser778 by Cdk5, followed by that of Ser774 by GSK3β[9,20]. Phosphorylation of Ser778 hampers the recruitment of Dynamin-1 by binding partners such as Endophilin[22], whereas phosphorylation of Ser774 inhibits Dynamin-1 activity[8]. These phosphorylations dampen CME but activate activity-dependent bulk endocytosis (ADBE), suggesting they might mediate the crosstalk between CME, FEME, and ADBE[8,9]. Several other endocytic proteins including Amphiphysin, are targets of these kinases in synapses[7,51]. Upon axon depolarization, the phosphatase Calcineurin is activated by the sudden Ca$^{2+}$ rise, mediating the acute dephosphorylation of Amphiphysin and Dynamin. Prompt removal of the inhibitory phosphorylations then swiftly activates compensatory endocytosis[52,53].

Here, we found that, similarly to their function of regulating endocytosis in synapses, Cdk5 and GSK3β suppress FEME (Figs. 1–4). Some of the mechanisms are shared, as for example the regulation of Dynamin-1, but others appear different in the case of FEME. Indeed, the kinases also regulate cargo protein sorting by Endophilin, as well as Dynein recruitment by Bin1 (Figs. 5–7). We furthermore established that Endophilin binds to PlexinA1 adaptor CRMP4 via a Proline-rich motif adjacent to the sites phosphorylated by Cdk5 and GSK3β, thereby placing the sorting of the receptor into FEME carriers under negative regulation by these kinases (Fig. 5). Our data also establish that, in mouse hippocampal neurons, axon extension and branching are regulated by the CRMP4-Endophilin interaction. These processes establish most of the synaptic connectivity in the central nervous system of vertebrates[32,54]. When CRMP4 was unlinked from the FEME pathway upon expression of the CRMP4-R525E mutant that cannot bind to Endophilin, axons were shorter and showed a defect in branching (Fig. 5). This was due to the reduced ability of these neurons to form functional AGCs or a higher tendency to collapse in response to low levels of endogenous Semaphorin 3A secreted into the media[55]. Although we cannot rule out a direct effect of the overexpression of the CRMP4 mutants, such hypersensitivity was likely induced by an increased level of endogenous Plexin A1 remaining at the cell surface upon blocking CRMP4 from being sorted into the FEME pathway. Conversely, overexpression of the CRMP4-S522A mutant, which blocks Cdk5-mediated inhibition of the CRMP4-Endophilin interaction induced the opposite phenotype: axon branching was normal and AGCs did not collapse, even upon ectopic addition of Semaphorin 3A (Fig. 5). This suggested that Plexin A1 was more actively internalised in these neurons, as we measured in HUVEC upon Cdk5 inhibition. Even though GSK3β does not appear to regulate the binding of CRMP4 to Endophilin, it controls other steps of FEME as well as axon branching and elongation through other functions of the CRMP complex[19,56]. Thus, our data shows that the dual regulation of FEME by Cdk5 and GSK3β has physiological importance in neurons, beyond the simple control of endocytosis.

We also found the uptake of Slit1 and its receptor ROBO1 to be under control of Cdk5 and GSK3β for their recruitment into FEME carriers (Suppl. Fig. 4). However, we do not know yet whether this regulatory mechanism is more generally used in the uptake of other FEME cargo as well. Several known FEME cargo proteins, including β1-Adrenergic receptor and CIN85 (the adaptor for EGFR), also contain a Serine–Proline motif (consensus site for Cdk5) within or adjacent to their Endophilin binding sites, suggesting that they could also be regulated by the mechanism described herein.

We have also shown that Cdk5 and GSK3β block the recruitment of Dynein onto FEME carriers (Fig. 6). We established that Bin1 engages the microtubule motor, and is another cytosolic marker of FEME carriers (Fig. 7). Thus, like Endophilin, Bin1 functions in FEME in addition to its role in CME. Ubiquitously expressed short isoforms 9 and 10 colocalized poorly with CME markers, in agreement with their lack of binding motifs to AP2 and Clathrin[50]. We cannot rule out that Bin1 has additional functions other than recruiting Dynein, nor can we exclude that another protein recruits the motor protein onto EPAs either in the absence of, or in parallel to Bin1. How Cdk5 and GSK3β control the loading of FEME carriers onto Dynein is not known at this point, as we failed to show an interaction between Bin1 and the obvious candidates p150glued and p27. However, multiple Dynein adaptors such as Ndel1L, Lis1, or BICD, have been reported to be phosphorylated by either kinases[37,57,58] and could be the link to Bin1. Additional layers of regulation are certainly at

play as both Cdk5 and GSK3β regulate Dynein processivity, in addition to cargo loading[37,38,59,60].

There may be additional levels of regulation by Cdk5 and GSK3β, either by regulation of other key steps of FEME or, more indirectly, by modulating the activity of other kinases. GSK3β inactivates both FAK1 (upon phosphorylation of Ser722[61]) and mTOR signaling (upon phosphorylation of TSC2 on Ser1341, Ser1337 and Ser1345[62]). Both FAK1 and mTORC1/2 were found to be FEME activators in our screen, as their acute inhibition blocked FEME (Fig. 1). This could imply that Cdk5 and GSK3β are master regulators of FEME acting both on the machinery as well as on other kinases potentially regulating the pathway. The finding that Endophilin binds to GSK3β suggest that local regulation is required to control FEME. The binding to GSK3β but not α stems from the presence of several PRMs in the former that are not conserved in GSK3α. Even though GSK3α and β share high sequence similarity, no clear role for GSK3α has been assigned in endocytosis, perhaps owing to a defect in local recruitment, due to its lack of binding to Endophilin.

It is now clear that various cell types have different levels of FEME activity. There are strong differences in the maximum number in FEME carriers produced upon growth factor addition: a factor of 7 between the weakest and the strongest tested (HEK293 and HUVEC, respectively, Suppl. Fig. 1). We also found that some cell types displayed spontaneous FEME (i.e., not induced experimentally), detected here in RPE1, hDFA and HUVEC cells. This is likely due to the high levels of growth factors in their culture in media: primary cells are routinely grown in medium supplemented with high doses of EGF, FGF, VEGF and/or IGF-1, which all trigger FEME[5]. Both Cdk5 and GSK3β control the level of FEME in a particular cell type, but only the sum of the activities of the two kinases (and perhaps that of other priming kinases that are yet to be identified) can predict the propensity of a cell type to be FEME active.

The signal for relieving the inhibition imposed by Cdk5 and GSK3β is not yet understood. The promptness of FEME activation upon stimulation of cargo receptors suggests that phosphatases likely erase inhibitory phosphorylations. However, which one could be acting downstream of receptors as diverse as $G\alpha_s$ or $G\alpha_i$-coupled GPCRs, RTKs, cytokine or cell guidance receptors (all the FEME cargoes known to date), is not obvious. In addition, kinases other than Cdk5 are likely priming GSK3β phosphorylation during FEME. Some Endophilin functions are regulated by LRKK2, DYRK1A and Src[63–65], but inhibitors toward these kinases did not affect spontaneous FEME in resting RPE1 cells. It is possible that dual inhibition together with GSK3β is required or that they regulate other FEME cargoes or processes. Thus, while a complex regulatory mechanism is likely to emerge from future work, the current study revealed the key role of Cdk5 and GSK3β in inhibiting FEME in absence of receptor activation.

## Methods

**Cell culture**. *Human* normal diploid hTERT-RPE-1 (ATCC CRL-4000, called RPE1 in this study) cells were cultured in DMEM:F12 HAM 1:1 v/v (Sigma D6421), 0.25% Sodium bicarbonate w/v (Sigma), 1 mM GlutaMAX-I (Thermo Fisher), 1× antibiotic–antimycotic (Thermo Fisher), and 10% fetal bovine serum (FBS; Thermo Fisher). *Human* Primary Dermal Fibroblasts (ATCC PCS-201-012, called hDFA in this study) were cultured in DMEM:F12 HAM 1:1 v/v (Sigma D6421), 7.5 mM GlutaMAX-I (Thermo Fisher), 1× antibiotic–antimycotic (Thermo Fisher), 2% FBS (Thermo Fisher), 0.8 μM Insulin (MP Biomedicals 0219390025), 10 ng/mL Basic Fibroblast Growth Factor (bFGF, LifeTech PHG0024), 50 μg/mL Hydrocortisone and 1 μg/mL ascorbic acid. *Human* umbilical vein endothelial cells, HUVEC (ATCC PCS-100-010 or a kind gift from Tom Nightingale (Queen Mary University)) were grown in endothelial cell growth medium containing 0.02 mL/mL Fetal Calf Serum, 5 ng/mL recombinant *human* Epidermal Growth Factor (EGF), 10 ng/mL Basic Fibroblast Growth Factor (bFGF), 20 ng/mL Insulin-like Growth Factor (IGF-1), 0.5 ng/mL recombinant *human* Vascular Endothelial Growth Factor 165 (VEGF-165), 1 μg/mL ascorbic acid, 22.5 μg/mL Heparin and 0.2 μg/mL Hydrocortisone (Promocell, C-22011).

*Human* HeLa cells (ATCC CCL-2), *human* embryonic kidney HEK293 (ATCC CRL-1573, called HEK in this study) and African green monkey BSC-1 (ECACC 85011422), were cultured in DMEM (Sigma D6546) supplemented with 10% FBS, 1 mM GlutaMAX-I (Thermo Fisher), 5% 1× antibiotic–antimycotic (Thermo Fisher). *Mouse* hippocampal neurons were initially seeded in neuronal attachment media (Minimum Essential Medium Eagle's with Earl's BSS (Sigma), 10% (w/v) FBS (Thermo Fisher), 1 mM sodium pyruvate (Sigma), 20% (w/v) glucose (Sigma), 2 mM glutamax (Thermo Fisher), antibiotic–antimycotic (Sigma)) and maintained at 37 °C, 5% $CO_2$ for 6 h after which the media was exchanged to neuronal maintenance media (Neurobasal (Thermo Fisher), 2% (w/v) B27 (Thermo Fisher), 2 mM GlutaMAX-I (Thermo Fisher), antibiotic-antimycotic (Sigma)). All the cells were maintained at 37 °C, 5% $CO_2$. Cells were regularly tested for mycoplasma contamination.

**Animals**. All procedures for the care and treatment of animals were in accordance with the Animals (Scientific procedures) Act 1986, and had full Home Office ethical approval. Animals were maintained under controlled conditions (temperature 20 ± 2 °C; 12 h light-dark cycle), were housed in conventional cages and had not been subject to previous procedures. Food and water were provided ad libitum. Wild-type C57BL6J mice were generated as a result of wild-type breeding; embryos of either sex were used for generating primary neuronal cultures.

**Neuronal culture and analysis**. Cultures of hippocampal neurons were prepared from wild-type C57BL6J *mouse* embryos (E16), using the following protocol[66]: mouse hippocampi were dissected from embryos brains in ice-cold HBSS (Gibco) supplemented withy 10 mM HEPES. Dissected hippocampi were incubated in the presence of 0.25% trypsin and 5 units/mL DNASe for 15 min at 37 °C, washed twice with HBSS with HEPES, and triturated to a single cell suspension in attachment media (MEM, (Gibco) containing 10% *horse* serum, 10 mM sodium pyruvate and 0.6% glucose) using a fire-polished glass pasteur pipette. Dissociated neurons were electroporated with CRMP4-EGFP constructs using Amaxa nucleofector (Lonza). This allowed tracing individual neurons/axons. Two hundred nanogram of DNA construct were used to electroporate $5 \times 10^4$ cells per condition. The neurons were then transferred into 96-well plates previously coated with poly-L-lysine (0.05 mg/mL, Sigma) in PBS overnight at 37 °C, 5% $CO_2$, followed by a second coating of Laminin (0.01 mg/mL) for 2 h at 37 °C, 5% $CO_2$. Approximately $5 \times 10^4$ transfected hippocampal neurons were seeded per well with an additional $9 \times 10^3$ non-transfected neurons seeded on top of the transfected neurons to support survival. Cells were initially allowed to attach in neuronal attachment media (described above) and grown at 37 °C, 5% $CO_2$ for 6 h, followed by media exchange with neuronal maintenance media. At day 4, cells were fixed and stained for F-actin (Phalloidin-555) and DNA (DAPI) and imaged using an imageXpress microscope (Molecular Devices). Axon length was determined by measuring the longest neurite using imageJ. In case an axon was branched the whole axon length was measured, including the branches. Axon branches were considered both arboreal (terminal) branches and also the interstitial (collateral) branches when they were located over or under 100 μm from the axon growth cones, respectively. Axon growth cone collapse was induced upon incubation of neuronal cultures with 5 nM recombinant semaphorin 3A for 20 min at 37 °C, 5% $CO_2$, before fixation with pre-warmed 4% PFA for 20 min at 37 °C and then stained. Growth cones were classified as collapsed if the actin lamellipodia/filopodia was shrunk or absent.

**Small compound inhibitors and ligands**. The following small compound inhibitors (amongst the best-reported inhibitors for each kinase[11,12]) were used: AZ191 (called DYRKi in this study Cayman 17693), AZD0530 aka Sacratinib (called SRCi in this study, Cayman 11497), BI-D1870 (called p90RSKi in this study, Cayman 15264), BIO-6-bromoindirubin-3′-oxime, aka BIO (called GSK3i2 in this study, (Sigma B1686), BI 2536 (called PLKi in this study, Selleckchem S1109), CDK1/2 inhibitor III (called Cdk1/2i in this study, Merck 217714), CHIR-99041 (called GSK3i1 in this study, Cayman 13122), Ciliobrevin D (called Ciliobrevin in this study, Calbiochem 250401), CX-4945 (called CK2i in this study, Cayman 16779), Dinaciclib (called Cdk1/2/5/9i in this study, MedChemExpress Hy-10492), Dobutamine (Sigma D0676), D4476 (called CK1i in this study, BioVision 1770), GDC-0879 (called BRAFi in this study, Tocris 4453), GDC-0941 (called PI3Ki in this study, Symansis SYG0941), Genistein (called Y-kinases in this study, Calbiochem 245834), GNE-7915 (called LRRK2i in this study, MedChemExpress Hy-10328), GSK2334470 (called PDKi in this study, Cayman 18095), GW 5074 (called CRAFi in this study, Santa Crux sc-200639), Harmine hydrochloride (called DYRKi in this study, Santa Crux sc2595136), JNK-IN-8 (called JNKi in this study MedChemExpress Hy-13319), KT 5720 (called PKAi in this study Cayman 10011011), MK2206 (called AKTi in this study, LKT Laboratories M4000), MLR 1023 (called LYNa in this study, Tocris 4582), PD0325901 (called MEKi in this study, Tocris 4192), PD0332991 aka Palbociclib (called Cdk4/6i in this study, Sigma PZ0199), PF-4708671 (called p70S6Ki in this study, MedChemExpress Hy-15773), PF-4800567 (called CK1Ei in this study, Cayman 19171), PHA-793887 (called Cdk2/5/7i in this study, ApexBio A5459), PND-1186 (called FAKi in this study, MedChemExpress Hy-13917), Purvalanol A (called Cdk1/2/4i in this study, Santa Cruz sc-224244), P505-15 (called SYKi in this study, Adooq Bioscence A11952), Roscovitine (called Cdk1/2/5i in this study, Santa Cruz sc-24002), RO-

3306 (called Cdk1i in this study, Cayman 15149), SCH772984 (called ERKi in this study, Selleckchem S7101), Staurosporine (called broad kinases in this study, Alomone Labs AM-2282), STO609 (called CaMKK1/2i in this study, Cayman 15325), TAK-632 (called panRAFi in this study, Selleckchem S7291), Torin 1 (called mTORC1i in this study, Tocris 4247), VX-745 (called p38i in this study, Med-ChemExpress Hy-10328) and ZM 447439 (called AurA/AurBi in this study, Cayman 13601). The following ligands were used: *human* Semaphorin 3A extracellular region 6 (6×N-terminal His-tag, R&D 1250-S3) and *human* Slit1 (6×C-terminal His-tag, R&D 6514-SL-050).

**Gene cloning and mutagenesis.** Full length and truncated genes (all *human*, unless specified) were amplified and cloned into pDONR201 (Invitrogen) and transferred into pEGFP, pTagRFP-T (called RFP in this study), pMyc, or pGEX-6P2 vectors converted into the Gateway system (pDEST vectors made from a pCI backbone), as appropriate: Endophilin-A2 (*SH3GL1*, IMAGE 3458016) full length and SH3 domain (aa 311-end); Endophilin-A1 (*SH3GL2 iso1*, FLJ 92732) full length and SH3 domain (aa 295-end); Endophilin-A3 (*SH3GL3* iso 1, IMAGE 5197246) full length and SH3 domain (aa 291-end); Bin1, also known as Amphiphysin-II (*BIN1 iso9*, cloned from *human* brain cDNA library) full length and SH3 domain (aa 366-end); full length CRMP2 (*DPYSL2*, DNASU HsCD00513405), full length CRMP3 (*DPYSL4*, NM_006426 Origene), full length *mouse* CRMP4 (*DPYSL3*, Origene 1197294), full length CRMP5 (*DPYSL5*, amplified from *human* brain cDNA library, Novagen), Ephrin receptor A1 cytoplasmic tail (aa 568-976) (*EPHA1*, DNASU HsCD00516390), Ephrin receptor A6 cytoplasmic tail (aa 572-1036) (*EPHA6*, DNASU HsCD00350501), Ephrin receptor B1 cytoplasmic tail (aa 259-346) (*EPHB1*, DNASU HsCD00038738), Ephrin receptor B4 cytoplasmic tail (aa 561-987) (*EPHB4*, DNASU HsCD00021508), Ephrin receptor B6 cytoplasmic tail (aa 616-1021) (*EPHB6*, DNASU HsCD00505529), Semaphorin 4D cytoplasmic tail (aa 756-862) (*SEMA4D*, amplified from *human* brain cDNA library, Novagen), Semaphorin 4F cytoplasmic tail (aa 681-770) (*SEMA4F*, DNASU HsCD00041427), Semaphorin 6A cytoplasmic tail (aa 671-1030) (*SEMA6A*, Sino Biologica HG11189-M), Semaphorin 6B cytoplasmic tail (aa 616-1021) (*SEMA6B*, amplified from *human* brain cDNA library, Novagen); Semaphorin 6D cytoplasmic tail (aa 684-1073) (*SEMA6D*, DNASU HsCD00516397), Plexin B1 cytoplasmic tail (aa 1512-2135) (*PLXNB1*, Addgene 25352), *mouse* Roundabout homolog 1 cytoplasmic tail (aa 880-1612) (*ROBO1*, DNASU HsCD00295416); Roundabout homolog 3 cytoplasmic tail (aa 912-1386) (*ROBO3*, DNASU HsCD00302878) and Netrin receptor UNC5B cytoplasmic tail (aa 398-945) (*UNC5B*, DNASU HsCD294959), EGFP-p27 (Addgene #15192); EGFP-p150Glued (Addgene #36154). *Bovine* Dynamin 1-EGFP and *rat* GST-Bin1 SH3 domain were kind gifts from Harvey McMahon (MRC Cambridge), EGFP-p50 dynamitin (full length *DCTN2*) was a kind gift from Serge Benichou (Institut Cochin, Paris) and EGFP-TPR (*mouse* KLC2 TPR domains aa 155–599) was a kind gift from Michael Way (Crick Institute, London). EGFP-tagged *human* full-length BAR domain proteins library was described before[6]. Point mutations P502A, S522A, S522D, R525E, and P526A were introduced in full length *mouse* CRMP4 and S774A and S778A were introduced in full length *rat* Dynamin-1 by site-directed mutagenesis and verified by sequencing. The list of primers used in this study is provided in Supplementary Table 1.

**Gene transfection.** For fixed cell colocalization experiments, cells seeded on 13 mm coverslips (placed in 24-well plates) were transfected using Lipofectamine 2000 (Thermo Fisher) or Nanofectin (PAA) and 10–500 ng DNA depending on the plasmids and the experiments (low or high overexpression). The levels of each plasmid were titrated down to low levels allowing good detection but limiting side effects of overexpression. Cells seeded onto live-cell imaging 35 mm glass bottom dishes (MatTek) were transfected using Lipofectamine 2000 (Thermo Fisher) and 50–250 ng DNA. For pull-down experiments, co-immunoprecipitation and EGFP-trap immunopurifications, HEK293 cells seeded in 6-well plates or 100 mm dishes were transfected using GeneJuice (Merck) and 1–3 μg DNA. Cells were incubated 16–24 h to express the constructs and were either imaged live, fixed (4% pre-warmed paraformaldehyde, 20 min at 37 °C) or processed to prepare cell extracts.

**siRNA suppression of gene expression.** The following siRNA oligos (all Stealth, Thermo Fisher) were used: Endophilin A1, A2, and A3 triple knock-down (Endo TKD) was achieved by combining oligos against Endophilin A1 (Thermo HSS109709; 2 oligos against *SH3GL2*), Endophilin A2 (Thermo HSS109707; 2 oligos against *SH3GL1*) and Endophilin A3 (Thermo HSS109712; 2 oligos against *human SH3GL3*); AP2 (Thermo HSS101955 (2 oligos against *human AP2M1*); CDK5 (Thermo HSS101729; 2 oligos against *human CDK5*); GSK3α/β double knock-down (DKD) was achieved by combining oligos against GSK3α (Thermo HSS104518; 2 oligos against *human GSK3A*) and GSK3β (Thermo HSS104522; 2 oligos against *human GSK3B*); CDK5+GSK3α/β triple knock-down (TKD) was achieved by combining aforementioned oligos against CDK5 and GSK3α/β; AMPH+Bin1 double knock-down (DKD) was achieved by combining oligos against Amphiphysin-1 (Thermo HSS100465; 2 oligos against *human AMPH*) and Bin1 (Thermo HSS100468; 2 oligos against *human BIN1*); FBP17+CIP4+TOCA-1 triple knock-down (TKD) was achieved by combining oligos against FBP17 (Thermo HSS118093; 2 oligos against *human FNBP1*),

CIP4 (Thermo HSS113814; 2 oligos against *human TRIP10*) and (Thermo HSS123422; 2 oligos against *human FNBP1L*); and Lamellipodin (Dharmacon ON-TARGETplus SMARTpool mix of J-031919-08, J-031919-07, J-031919-06, and J-031919-05 targeting *human RAPH1*). Control siRNA used were Invitrogen Stealth control (scrambled) oligo 138782. Cells seeded on 13 mm coverslips placed in 24-well plates were transfected twice (on day 1 and 2) with Oligofectamine or RNAi MAX (Thermo Fisher) complexed with 20 pmol of each indicated siRNA and analyzed 3–4 days after the first transfection. RNAi knock-down efficiency was verified by western-blotting or immunofluorescence counter-staining. The use of validated pools of siRNA targeting the same genes increased the knock-down efficiency and specificity.

**Antibodies.** The following primary antibodies were used for immunostaining or immunoblotting: anti-EGFP ab290 (*rabbit* polyclonal, Abcam290) used at 0.5 μg/mL (1:8000 dilution), anti-EGFP clones 7.1 and 13.1 (*mouse* monoclonal, Roche 11814460001) used at 0.5 μg/mL (1:1000 dilution), anti-Endophilin A2 clone H-60 (*rabbit* polyclonal, Santa Cruz 25495)), used at 1 μg/mL (1:100 dilution), anti-Endophilin A2 clone A-11 (*mouse* polyclonal, Santa Cruz 365704) used at 1 μg/mL (1:200 dilution), anti-β1 adrenergic receptor (*rabbit* polyclonal, Abcam ab3442) used at 1μg/mL (1:1000 dilution), anti-CRMP4 (*rabbit* polyclonal, Milipore 5454) used at 2 μg/mL (1:100 dilution), anti-Dynein clone 74.1 (*mouse* monoclonal, eBioscience 14-9772-80) used at 2 μg/mL (1:100 dilution), anti-Plexin A1 (*rabbit* polyclonal recognizing the ectodomain of PlexinA1, Abcam Ab32960) used at 1 μg/mL (1:100 dilution), anti-ROBO1 (sheep polyclonal, AF7118 R&D Systems) used at 1 μg/mL (1:500 dilution), anti-LAMP-1 (*mouse* monoclonal clone H4A3-c, Developmental Studies Hybridoma Bank) used at 1:5000 dilution, anti-phosphorylated Ser9 GSK3β clone D85E12 (*rabbit* monoclonal, Cell Signaling Technology 5558) used at 1 μg/mL (1:400 dilution), anti-GSKα/β D75D3 (*rabbit* polyclonal, Cell Signaling Technology 5676) used at 1 μg/mL (1:400 dilution), anti-Dynamin 1 clone 41 (*mouse* monoclonal, BD Pharmigen 610245) used at 1 μg/mL (1:250 dilution), anti-Bin1 (*rabbit* polyclonal, GeneTex GTX103259) used at 2 μg/mL (1:500 dilution), anti-Lamellipodin (*rabbit* polyclonal, Atlas Antibodies HPA020027) used at 2 μg/mL (1:50 dilution), anti-CIP4, (*mouse* monoclonal clone 21, Santa Cruz sc-135868) used at 1 μg/mL (1:200 dilution), anti-α Tubulin clone TUB2.1 (*mouse* monoclonal, Abcam ab11308) used at 0.5 μg/mL (1:2000 dilution), and anti-His tag clone D3I10 (*rabbit* polyclonal, Cell Signaling Technology 12698) used at 1 μg/mL (1:400 dilution). The following secondary antibodies were used for microscopy: Alexa Fluor 488 *goat* anti-*mouse* IgG (Thermo Scientific A-11001), Alexa Fluor 555 *goat* anti-*mouse* IgG, (Thermo Scientific A-21422), Alexa Fluor 488 *goat* anti-*rabbit* IgG, (Thermo Scientific A-11008), Alexa Fluor 555 *goat* anti-*rabbit* IgG, (Thermo Scientific A-21428), Alexa Fluor 388 *donkey* anti-*Sheep* IgG (Thermo Scientific A-11015), Alexa Fluor 555 *donkey* anti-*mouse* IgG (Thermo Scientific A-31570), and for immunoblot; *goat* anti-*mouse* IgG-HRP conjugated (Bio-Rad 1706516) and *goat* anti-*rabbit* IgG-HRP conjugated (Bio-Rad 1706519). Actin was stained using Phalloidin-Alexa647 (Cell Signaling Technology 8940) and DNA using DRAQ5 (BioStatus DR50200).

**Cell stimulation and cargo uptake.** Cells were kept at 37 °C and 5% $CO_2$ during the whole assay (apart during medium exchanges) and never serum-starved or pre-incubated at 4 °C. Resting conditions correspond to cells being cultured in 10% serum media and directly fixed (4% pre-warmed paraformaldehyde) for 20 min at 37 °C. Kinase inhibition was achieved by incubating cells grow in full medium (10% serum) with the indicated small compound inhibitors at the indicated concentrations and for the indicated times at 37 °C before being washed once with 37 °C pre-warmed PBS and fixed (4% pre-warmed paraformaldehyde) for 20 min at 37 °C. Serum stimulation was achieved by adding 37 °C pre-warmed 10% serum on complete medium (20% serum final) for the indicated times. β1 adrenergic receptor stimulation (which activates FEME) was performed by incubating cells at 37 °C for 4 or 30 min with pre-warmed medium containing 10μM dobutamine. Plexin A1 uptake was performed by incubating cells at 37 °C for 5–20 min with pre-warmed medium containing 20 nM Semaphorin 3A and 10 μg/mL anti-PlexinA1 antibodies (recognizing the ectodomain of PlexinA1). ROBO1 stimulation was performed by incubating cells at 37 °C for 10 min with pre-warmed medium containing 2 nM Slit1-(His)₆. In some experiments, cells were pre-incubated at 37 °C for the indicated times with small compound inhibitors before stimulation with dobutamine, Semaphorin 3A or Slit1 (in constant inhibitor concentration). After the incubation periods at 37 °C, cells stimulated as described above were quickly washed once with 37 °C pre-warmed PBS to removed unbound ligands and fixed with pre-warmed 4% PFA for 20 min at 37 °C (to preserve Endophilin staining and FEME carriers morphology). In some experiments, unbound and cell suface anti-PlexinA1 antibodies were removed by one quick wash in ice-cold PBS++ (containing 1 mM $CaCl_2$ and 1 mM $MgCl_2$) followed by two 5 min incubations on ice in acid stripping buffer (150 mM NaCl, 5 mM KCl, 1 mM $CaCl_2$, 1 mM $MgCl_2$, 0.2 M acetic acid adjusted to pH 2.5), followed by two washes on ice in PBS++ to normalize pH back to 7. Fixed cells were then washed three times with PBS and one time with PBS supplemented with 50 mM $NH_4Cl$ to quench free PFA. Cells were then permeabilized (0.05% saponin), immunostained and imaged as described below.

**Immunostaining and confocal fluorescence microscopy.** Cells were fixed with 4% PFA at 37 °C for 20 min, washed three times with PBS and one time with PBS

with 50 mM NH$_4$Cl to quench free PFA. Cells were then permeabilized for 5 min with PBS with 0.05% saponin and immunostained with primary and secondary antibodies in PBS with 0.05% saponin (Sigma) and 5% heat inactivated Horse Serum. Cover slips 0.13–0.16 mm (Academy) were mounted on slides (Thermo scientific) using immunomount DAPCO (GeneTex) and imaged using a laser scanning confocal microscope (TCS Sp5 AOBS; Leica) equipped with a 63× objective. For Alexa488, the illumination was at 488 nm and emission collected between 498 and 548 nm; for Alexa555 the laser illumination was at 543 nm and emission collected between 555 and 620 nm; for Alexa647 and DRAQ5, the laser illumination was at 633 nm and emission collected between 660 and 746 nm. Correlation between total Cdk5 or GSK3β cellular levels and number of EPAs was determined from single cell measurements on the same cells (i.e., matching Cdk5 or GSK3β levels with the number of EPAs in each individual cell measured). The percentages of endophilin spots located at the leading edge of cells or FEME carriers (EPAs) positive for endogenous GSK3β, Dynamin-1, CRMP4, Dynein, or Bin1 were determined by line scans using Volocity 6.0. Colocalization of over-expressed Endophilin-A2-RFP and EGFP-tagged CRMP4 constructs were determined by Manders' overlap using Volocity 6.0. The percentages of FEME carriers (EPAs) positive for BAR domain tagged with EGFP were determined by line scans using Volocity 6.0. The percentages of Bin1 spots positive for endogenous Endophilin, Lamellipodin, CIP4, or Dynein were determined by line scans using Volocity 6.0. Levels of endogenous β1AR, PlexinA1, or recombinant Slit1 internalized into FEME carriers were measured by using Volocity 6.0. Micrographs were cropped using ImageJ 1.51s (NIH) or Photoshop CS5 (Adobe).

**Live-cell confocal fluorescent microscopy.** Just before live-cell imaging, the medium of cells grown on MatTek dishes was changed to α-MEM without phenol red, supplemented with 20 mM HEPES, pH 7.4 and 5% FBS and placed into a temperature controlled chamber on the microscope stage with 95% air: 5% CO$_2$ and 100% humidity. Live-cell imaging data were acquired using a fully motorized inverted microscope (Eclipse TE-2000, Nikon) equipped with a CSU-X1 spinning disk confocal head (UltraVIEW VoX, Perkin-Elmer) using a 60× lens (Plan Apochromat VC, 1.4 NA, Nikon) under control of Volocity 6.0 (Improvision, Perkin-Elmer). 14-bit digital images were obtained with a cooled EMCCD camera (9100-02, Hamamatsu, Japan). Four 50 mW solid-state lasers (405, 488, 561, and 647 nm; Crystal Laser and Melles Griots) coupled to individual acoustic-optical tunable filter (AOTF) were used as light source to excite EGFP and TagRFP-T. Rapid two-colour time-lapses were acquired at 500 ms to 2 s intervals, using a dual (525/50; 640/120, Chroma) emission filter, respectively. The power of the lasers supported excitation times of 50 ms in each wavelength and the AOTFs allowed minimum delay (~1 ms) between two colors (e.g., delay between green-red for each timepoint), which was an important factor to assess the colocalization between markers.

**Protein purification and pull down experiments.** GST or GST-tagged SH3 domains were expressed in BL21 (DE3) *E. coli* (New England Biolabs). Cells were lysed by sonication in presence of lysozyme (Affymetrix), protease inhibitor (Thermo Scientific), and DNAse powder (Sigma-Aldrich), spun at 5,000×g for 1 h at 4 °C. The supernatants containing the GST or GST-SH3 domains (soluble fraction) were concentrated (to ~60 mg/mL) using a Centricon Plus 70-1000 NMWL (Centricon) for 1 h at 4 °C and then incubated rotating with GST-sepharose beads (PierceTM glutathione superflow agarose) overnight at 4 °C. The beads were washed ten times with ice-cold PBS and kept in PBS and sodium azide 0.02% solution at 4 °C and used in pull-down assays. Cell lysates—non-transfected or overexpressing EGFP-tagged proteins—were prepared in lysis buffer (20 mM HEPES, 1 mM EDTA, 0.2% Triton X-100 and protease inhibitor cocktail (Roche)) briefly sonicated (three times 5 s pulses with 30 s rest, 10 μm amplitude) and spun at 20,000 × g for 10 min at 4 °C. Cell lysates were incubated with bead-bound proteins (amounts were calibrated by gel electrophoresis followed by Coomasie to equivalent amounts) overnight at 4 °C and then centrifuged at 7500 × g and washed three times with lysis buffer. The remaining bead pellet was boiled in SDS sample buffer and run on SDS-PAGE. Input lanes correspond to 5% of cell extract. The final pellets (bound fractions), supernatants (unbound fractions) and original extracts (input fractions) were boiled in sample buffer and ran on SDS-PAGE. The proteins were transferred onto PVDF membrane and immunoblotted using anti-EGFP antibodies or antibodies against endogenous proteins, as indicated, followed by HRP-coupled secondary antibodies (BioRad). Blots were developed with the ECL kit (Thermo Fischer Scientific or Merck Millipore) and x-ray film and quantified using ImageLab 6.1 (Bio-Rad).

**Co-immunoprecipitations.** HEK293 cells were co-transfected with equal amounts (1–3 μg) of Myc-tagged and EGFP-tagged BAR domain constructs. After 16–24 h expression, cells were quickly washed with cold PBS, lysed in ice-cold lysis buffer (10 mM Tris HCL pH 7.5, 150 mM NaCl, 0.5 mM EDTA, 0.5% NP40 and a protease and phosphatase inhibitor cocktail (Thermo Scientific)) and spun at 14,000 × g for 10 min at 4 °C. Cell lysates were incubated with GFP-TRAP_A or M (Chromotek) bead slurry for 1–16 h at 4 °C. The beads were washed three times (10 mM Tris HCL pH 7.5, 150 mM NaCl, 0.5 mM EDTA). The final pellets and unbound fractions were boiled in SDS sample buffer and ran on SDS-PAGE (input lanes

correspond to 1–10% of cell extracts). The proteins were transferred onto PVDF membrane and immunoblotted using anti-Myc, anti-EGFP, antibodies, or antibodies against endogenous proteins, as indicated, followed by HRP-coupled secondary antibodies (BioRad). Blots were developed with the ECL kit (Thermo Fischer Scientific or Merck Millipore) and x-ray films. Uncropped and unprocessed Western blots are shown in the Source Data file.

**FEME carrier isolation.** RPE1 cells grown on 15 cm dishes were stimulated with extra 10% FBS (20% final) for 10 min at 37 °C, quickly rinsed with ice-cold PBS$^{+++}$ (PBS with 1 mM Ca$^{2+}$, protease and phosphatase inhhibitors), collected using a cell scraper and pelleted (400 × g, 5 min, 4 °C). Cell pellets were loosened in 1 mL of ice-cold homogenization buffer A (HBA) (3 mM Imidazol pH 7.4, 1 mM EDTA, and 0.03 mM cycloheximide plus protease and phosphatase inhibitor cocktail), spun (1300 × g, 10 min, 4 °C), resuspended into one volume of HBA and incubated for 20 min on ice. One volume of homogenization buffer B (HBB; HBA containing 500 mM sucrose) was added and cells were mechanically lysed through 25G needles, avoiding nuclei disruption. Homogenates were diluted into HBB (one part homogenate and 0.7 part HBB) and post-nuclear supernatants (PNS) were collected after spinning (two time, 2000 × g, 10 min, 4 °C). Sucrose concentration of the PNS was adjusted to 40.6% using 62% sucrose solution (2.351 M sucrose, 3 mM Imidazole pH 7.4) and 1 volume was loaded at the bottom of ultracentrifuge tubes. Cushions of 35% sucrose (1.5 volume), 25% (1 volume) and 8% sucrose (1 volume) were carefully added and the tubes were centrifuged at 210,000 × g for 3 h at 4 °C. Gradients were divided in ten fractions and used either for immunoblotting or immunopurification. Immunoblotting was performed after sucrose gradient fractions were concentrated following protein precipitation (25%/v TCA, incubated for 10 min at 4 °C, spun at 20,000 × g for 5 min at 4 °C; pellets were washed twice with −20 °C pre-chilled acetone and air dried before resuspension in SDS sample buffer). Selected sucrose gradient fractions were submitted to immunoprecipitation using an anti-Endophilin antibody (mouse IgG2a/κ-ligh chain clone A-11, sc365704) coupled directly to hydrazide-terminated magnetic beads (Bioclone Inc; coupling performed following manufacturer's instructions) to allow for elution of the binding material without denaturation. Selected sucrose gradient fractions were adjusted to 500 μL with HBA and incubated (overnight at 4 °C, slow rotation) with 1:50th volume of anti-Endophilin coupled magnetic beads (pre-equilibrated in HBA with 250 mM sucrose). Unbound material was isolated, and the beads washed three times in 1 mL ice-cold HBB (washes were also kept for analysis). Bound material was released in 500 mL elution buffer (0.1 M glycine pH 2, 250 mM sucrose) for 10 min (4 °C, slow rotation), neutralized (>200 μL of 1 M Tris pH 8 until pH back to neutral) and prepared for immunoblotting. Lipids were analyzed by SDS-PAGE using Bis-Tris gels run in MES buffer (to avoid excess counterions at the gel front that interfere with lipid staining), and stained using alcohol-free coomassie (0.1% Coomassie in 10% acetic acid for 5 min and destained in water; alcohol was omited to not solubilize the lipids out of the gels), as established previously[45].

**Immunoprecipitation for MS experiments.** RPE1 cells were grown on 10 cm dishes to a confluence of 80% before harvest. For stimulated condition, cell media was supplemented with extra 10% FBS (20% final) for 5 min at 37 °C. For resting condition, no additional treatment was performed prior to cell lysis. Cells were washed two times in ice-cold PBS before being gently scraped into lysis buffer (10 mM Tris HCL pH 7.5, 150 mM NaCl, 0.5 mM EDTA, 0.5% NP40 and a protease and phosphatase inhibitor cocktail (Thermo Scientific)) and incubated on ice for 30 min, then centrifuged at 17,000×g for 10 min. Anti-endophilin antibodies (Endophilin II A-11) were coupled in-house to hydrazide-terminated magnetic beads (H-beads). Ten microliter of H-beads pre-washed in lysis buffer were incubated with 470 μL of cell lysate overnight at 4 °C with end-over-end rotation. The beads were washed three times in lysis buffer then boiled in 50 μL SDS sample buffer for 10 min.

**MS sample preparation by in-gel digestion.** Samples were separated by SDS-PAGE on a 4–12% gel and stained with InstantBlue (Expedeon) staining solution. Sample lanes were cut into ten sections, then further cut into 1 mm$^3$ pieces and washed in destaining solution (40% ethanol, 10% glacial acetic acid in water). Proteins were reduced in 10 mM DTT, then alkylated with 20 mM iodoacetamide in 50 mM ammonium bicarbonate buffers. Gel pieces were washed then immersed in a 10 ng/μL Trypsin buffer in 50 mM ammonium bicarbonate and digested for 16–18 h at 37 °C. Gel pieces were incubated in Elution buffer (1% formic acid, 2% acetonitrile in LC-MS grade water (Thermo Scientific)) and dried in a SpeedVac. Peptides resuspended in 0.5% acetic acid in water and desalted on C18-Stagetips, then dried in a SpeedVac. Peptides were resuspended in LC-MS running buffer (3% acetonitrile, 0.1% formic acid in water) prior to analysis by LC-MS and spiked with *E. coli* ClpB peptides (Waters, UK) such that 50 fmol of spiked-in peptide standard was introduced per injection.

**Liquid chromatography and mass spectrometry data acquisition.** LC-MS grade solvents were used for all chromatographic steps. Separation of peptides was performed using a Waters NanoAcquity Ultra-Performance Liquid Chromatography system. Peptides were reconstituted in 97:3 H2O:acetonitrile + 0.1% formic acid. The mobile phase was: A) H2O + 0.1% formic acid and B) Acetonitrile +

0.1% formic acid. Desalting of the samples was performed online using a reversed-phase C18 trapping column (180 μm internal diameter, 20 mm length, 5 μm particle size; Waters). Peptides were separated by a linear gradient (0.3 μL/min, 35 °C column temperature; 97–60% Buffer A over 60 min) using an Acquity UPLC M-Class Peptide BEH C18 column (130 Å pore size, 75 μm internal diameter, 250 mm length, 1.7 μm particle size, Waters, UK). [Glu1]-fibrinopeptide B (GFP, Waters, UK) was used as lockmass at 100 fmol/μL. Lockmass solution was delivered from an auxiliary pump operating at 0.5 μL/min to a reference sprayer sampled every 60 s. The nanoLC was coupled online through a nanoflow sprayer to a QToF hybrid mass spectrometer (Synapt G2-Si; Waters, UK). Accurate mass measurements were made using a data-independent mode of acquisition (HDMS$^E$). Each sample was analyzed in technical duplicate.

**Database searching and MS analysis**. Raw data was analyzed using Progenesis v4.0 (Waters, UK). Data were queried against a Homo sapiens FASTA protein database (UniProt proteome:UP000005640) concatenated with a list of common contaminants obtained from the Global Proteome Machine (ftp://ftp.thegpm.org/fasta/cRAP) and E. coli ClpB, which acted as a standard for label-free absolute protein quantitation[67]. Carbamidomethyl-C was specified as a fixed modification and Oxidation (M) and Phosphorylation of STY were specified as variable modifications. A maximum of two missed cleavages were tolerated in the analysis to account for incomplete digestion. For peptide identification three corresponding fragment ions were set as a minimum criterion whereas for protein identification a minimum of seven fragment ions were required. Protein false discovery rate was set at 1%. Samples were normalized to Endophilin peptide abundance and fractions were combined in-silico in Progenesis to obtain absolute protein abundances for differential expression analysis.

**Experimental design**. A strategy for randomization, stratification, or blind selection of samples has not been carried out. Sample sizes were not chosen based on pre-specified effect size. Instead, multiple independent experiments were carried out using several independent biological replicates as detailed in the figure legends.

**Quantification and statistical analysis**. All experiments were repeated at least three times, giving similar results. For all figures, results shown are mean ± standard error of the mean (SEM). Statistical testing was performed using Excel 14.7.3 (Microsoft) and Prism 6 (GraphPad Software). Comparisons of data were performed by one-way analysis of variance (ANOVA) with Tukey's multiple comparison test or by two-way ANOVA with Tukey's multiple comparisons test, as appropriated. NS, non significant ($P > 0.05$); *$P < 0.05$, **$P < 0.01$, ***$P < 0.001$.

**Reporting summary**. Further information on research design is available in the Nature Research Reporting Summary linked to this article.

## Data availability

LC-MS/MS data have been deposited in ProteomeXchange Consortium via the PRIDE partner repository with identifier PXD021138. All other data that support the findings of this study are available from the corresponding author upon reasonable request. Source data are provided with this paper.

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

## Acknowledgements
We thank Mina Edwards and Marta Martins for technical help, Alexandra Chittka (University College London), Tom Nightingale (Queen Mary University), Harvey McMahon (MRC Cambridge), Serge Benichou (Institut Cochin, Paris) and Michael Way (Crick Institute, London) for the kind gift of reagents and the members of the Boucrot lab for helpful comments. A.P.A.F. was supported by the Fundação para a Ciência e Tecnologia. A.C. was supported by a Biotechnology and Biological Sciences Research Council (BBSRC) LIDo PhD scholarship. S.S. was supported by a Medical Research Council PhD scholarship. K.S. was a recipient of summer internship from the Lister Institute of Preventive Medicine. E.F.H. was the recipient of a Marie Skłodowska-Curie grant (661733). J.T.K. received support from an ERC starting grant (282430; Fuelling synapses) and from the MRC (MR/N025644/1). The G2-Si ion mobility mass spectrometer was purchased with a grant from the Wellcome Trust (104913/Z/14/ZBM) to K.T.; and E.B. was a BBSRC David Phillips Research Fellow (BB/R01551X), a Lister Institute Research Fellow and a recipient of a BBSRC Pathfinder grant (BB/R01552X) and of a Birkbeck/Wellcome Trust Institutional Strategic Support Fund (ISSF2) Career Development Award.

## Author contributions
A.P.A.F., A.C., S.C.R., J.P., K.S., and E.B. performed biochemical assays; A.P.A.F., A.C., S.C.R., E.F.H., and E.B. performed cell biology experiments; A.P.A.F., E.F.H., and A.C. performed neuron experiments, supervised by J.T.K. and E.B.; S.S. performed mass spectrometry, under the supervision of K.T.; D.M. designed and performed FEME carrier isolation; A.P.A.F. and L.C.W.H. generated critical reagents; A.P.A.F., A.C., S.C.R., K. McG., and E.B. performed image acquisition and analysis; E.B. designed the research and supervised the project. E.B. wrote the manuscript with input from all the other authors.

## Competing interests
The authors declare no competing interests.
