## [Peer Review File · Nature Communications]

Reviewers' comments:

Reviewer #1 (Remarks to the Author):

How fast endophilin-mediated endocytosis (FEME) is regulated is an important question. In this study, the authors reasoned that since FEME is inhibited in cells starved of serum, protein kinases involved in growth factor signaling might regulate FEME activity. Using small molecule inhibitors to screen for kinases that either activate or inhibit endophilin-positive assembly (EPA) formation, the authors identified Cdk5 and GSK3 as negative regulators of FEME. They provide evidence that Cdk5 inhibits Semaphorin 3A-stimulated Plexin A1 uptake via blocking CRMP4 recruitment onto endophilin-labeled FEME carriers. They also show that ROBO1 colocalization with endophilin-labeled structures increases upon GSK3 inhibition. The authors further show the effect of kinase inhibition on recruitment of dynamin and dynein onto endophilin-labeled FEME carriers, and conclude that Cdk5 and GSK3 antagonize FEME budding by inhibiting dynamin and dynein function. The identification of Cdk5 and GSK3 as negative regulators for FEME is an important finding for understanding the regulatory mechanisms underlying endophilin-dependent, clathrin-independent endocytosis. However, more evidence is needed to support the conclusion that ligand-induced uptake of Plexin A1 and ROBO1 and their downstream signaling are mediated by FEME and regulated by Cdk5 and/or GSK3. Besides, some of the experiments and conclusions are similar to those previously reported (Renard H-F et al., *Nature* 517, 493-496, 2015), unless the endophilin-dependent, clathrin-independent endocytosis pathways of different cargoes identified in the papers (*Nature* 517, 493-496 and 460-465, and this study) require totally different mechanisms.

Major points:

1. The authors observed that some cell types display robust FEME while others do not. If Cdk5 and GSK3 indeed play critical roles in controlling FEME activity, as supporting evidence, the authors should compare the activity levels of Cdk5 and GSK3 in cell lines growing in normal medium with 10% serum, e.g. HUVEC and RPE1 vs HeLa, HEK293 or BSC1.
2. Figure 2E-G: It was reported that Semaphorin 3A-induced receptor endocytosis is mediated by L1 CAM and is AP2-dependent (Castellani V et al., *Mol Cell Neurosci* 26: 89-100, 2014), whereas FEME is AP2-independent. To determine whether or not it is endophilin-dependent and clathrin-independent, the authors need to test the effect of endophilin depletion and CME inhibition on internalization of Plexin A1 upon semaphoring 3A treatment.
3. Figure 2G: the authors claim that the longer neurite phenotypes in Figure 2H and cell collapse in Figure 2I are likely mediated by the increased levels of Plexin A1, caused by FEME inhibition or uncoupling of CRMP4 to endophilin. If it is true, there should be more plexin A1 signals on the cell surface (detect either by immunostaining or surface protein biotinylation) in EEN TKO cells or cells overexpressing CRMP4 mutants. Rather than speculating, the authors should provide experimental data to prove it.
4. In Figure 2H, the authors used neurite outgrowth of PC12 cells as readout to assess the role of Plexin A1 uptake in FEME carriers, and found that overexpression of endophilin-binding deficient mutants of CRMP4 caused increase in neurite length. However, as it is well known that NGF is the major growth factor that induces neurite outgrowth of PC12, the authors should either use a cellular model for semaphoring 3A signaling or clarify whether the phenotype is caused by defect in plexin A1 uptake or in endophilin-mediated FEME of the NGF-TrKA signal complex.

5. Figure 3B: the experimental group of 2 nM Slit1-treated cells is missing.
6. Figure 3C: 1) The signal intensity of Slit1 in AP2 KD cells looks much weaker than control, which is inconsistent with the quantification results in the right panel. How many cells were analyzed? 2) In figure legends the authors said it is “Fluorescently labeled Slit1 uptake”, whereas in METHODS it was described as “...cells were stained with antibodies or phalloidin...”. Please clarify which labeling method was used in the Slit uptake assay. 3) If GSK3 indeed inhibits Slit-induced Robo uptake, treatment of cells with GSK3 inhibitor should also cause an increase in Slit1 uptake that is opposite to the phenotype caused by Endo TKD.
7. Figure 3D: again there is inconsistency in the figure legend and METHODS. The cells were cortical neurons according to the legend. In METHODS they were described as hippocampal neurons. Furthermore, did the authors determine the KD efficiency of EEN1, 2 and 3 in neurons? To measure axon extension, the authors should take pictures of whole cells and measure the length of axons of individual neurons (at least 30 neurons/group), not Tau total signal intensity of the view field, which most likely represents cell density, rather than the extent of axon outgrowth.
8. Figure 4. The data showing that kinesin is not involved in betaAR1 uptake is very nice. However, the experiments designed to determine whether the dynein motor complex and microtubules are involved in FEME carrier budding and corresponding conclusions are similar to those reported by Renard et al. in their study of endophilin A2-mediated Shiga Toxin B endocytosis (Renard H-F et al., Nature 517, 493-496, 2015). It would be better to say these data (in Figure 4-D) are in agreement with previous study and cite the paper.
9. The authors showed that inhibition of Cdk5 and GSK3 increased localization of dynein to endophilin-labeled structures, and stated “this is consistent with the regulation of cargo-binding complexes loading onto Dynein by Cdk5 and GSK3 kinases”. However, it was reported that Cdk5 phosphorylation of the dynein regulator Ndel1 stimulates cargo transport by promoting dynein-microtubule interaction (J Neurosci 31, 17207-17219, 2011 and Traffic 18, 808-824, 2017). The increased colocalization between the motor and cargo carriers might be a result of failure of the motor to move on microtubule tracks, hence the accumulation of dynein-associated EPAs rather than a cargo loading defect. Therefore, the authors should interpret the data more carefully to explain this apparent discrepancy. A motor-cargo interaction assay (e.g. vesicle IP using antibodies against endophilin or the transmembrane receptor cargoes) is a good approach to determine whether or not the kinases regulate cargo loading onto dynein. It would also help if the authors test whether kinase inhibition affects the distribution of FEME carriers to microtubules by immunostaining (similar to Figure 4D).

Minor points:

1. Figure 2E, top left label: “anti-Endophilin” should be “merged” or “Endo EGFP-CRMP”?
2. Supplementary Figure S1C, top left label “anti-Endophilin” should be “merged” or “Endo EGFP”? Similarly, in panel D, the images labeled with “EndophilinA2-RFP” are merged images of RFP and GFP.
3. Supplementary Figure S1D, how was colocalization quantified? Pearson coefficient or Mander’s overlap?
4. In “QUANTIFICATION AND STATISTICAL ANALYSIS”, the authors stated that “P values < 0.01 were considered significant”, whereas in Figure 1B, *p < 0.05 is considered significant. Please be consistent.

Reviewer #3 (Remarks to the Author):

The work by Ferreira et al., performed a kinase screen for the fast endophilin-mediated endocytosis (FEME) using an imaging-based assay. They then focused on the negative regulator of the FEME, namely Cdk5 and GSK3. To demonstrate how Cdk5 blocks FEME, the authors turned to a known interactor of Endophilin SH3 domain, CRMP4, and showed that phosphorylation-dead or -mimetic mutants have opposite effect on its interaction with endophilin. For the effect of GSK2, receptor sorting of ROBO1 was shown to be affected. Lastly, the combination of Cdk5 and GSK3 inhibition affected recruitment of dynamin and dynein.

Overall I think characterizing the drug sensitivity of FEME pathway could be of interest to readers interested in this pathway. This approach can also provide some quick entry points for new insights. However, these initial insights have to be followed up with more rigorous genetic experiments to be truly useful. In comparison to the previous work by the authors, the current study provided incremental rather than conceptual advances, with - I am sorry to say this - less solid evidence. While I think many of the additional experiments are feasible and I am positive that the authors have considered them as well, I am not sure whether they could be incorporated in a revised draft because many authors appear to have left.

The major concern I have is the sole use of endophilin immunofluorescence signals as readout for the FEME pathway. Endophilin does participate in clathrin-mediated endocytosis (CME). The effect of these chemical inhibitors could be pleiotropic because many of their substrates also work in CME. Lastly, the recruitment of endophilin to the plasma membrane and its duration on the membrane could be independent from the actual uptake (For instance, the amount of clathrin accumulated on the membrane is most likely not an accurate readout for the efficiency of CME). Therefore, it is rather surprising that the authors did not provide evidence that these inhibitors do affect the uptake of the receptors, either positively or negatively, and whether such effects are clathrin-independent. For example, line 112, "Increased levels of Plexin A1 at the cell surface": was it shown? Does the number of endophilin dots positively correlate with endocytosis of these receptors? In the original 2015 paper, a number of assays were used for this purpose, so it should be technically feasible. In addition, how the scoring system worked was somewhat obscure. Was it based on automatic quantitation or visual impressions? It was not mentioned in the method.

Other comments:

1. The recruitment of EGFP-tagged CRMP4 or R525E onto FEME carriers in Fig 2 is rather difficult to appreciate. Also the data showing the increased colocalization upon Cdk5 inhibitor treatment requires some faith to believe in.
2. Do endophilin and ROBO1 really colocalize? The two signals do not seem to be in the same compartments.
3. The involvement of CRMP4 was tested by biochemical experiments and dominant-negative mutants. To demonstrate that it is the key substrate of Cdk5 that affects FEME, some loss-of-

function experiments are needed.

4. Fig 3c needs to show the GSK3 inhibitor under these conditions. AP2 KD seems to reduce Slit1 internalization in the image shown, which seems to contradict with the statement in the text.
5. How does GSK regulate the sorting of ROBO1?
6. Line 134 and the following paragraph, does the author mean “fission” instead of “budding”?
7. Are dynein recruited to the late stage of FEME similar to Fig 1F?
8. Line 80 has unknown characters.
9. Line 108 has missing reference.

Reviewer #4 (Remarks to the Author):

The manuscript by Ferreira et al. identified Cdk5 and GSK3 as negative regulators of fast endophilin-mediated endocytosis (FEME). Specifically, by using small molecule inhibitors, the study showed that specific inhibitors of Cdk5 or GSK3 were sufficient to activate FEME or induce the production of endocytic carriers containing membrane receptors. In addition, the study found that Cdk5 and GSK3, with Cdk5 as the priming kinase, acted to phosphorylate CRMP4 and its subsequent dissociation from endophilin and formation of FEME carrying axon repulsion cue semaphoring receptor Plexin A1. As a result, phosphomimetic CRMP4, which could not bind to endophilin and activated FEME carrying Plexin A1, led to enhanced neurite outgrowth of PC12 cells. Furthermore, the study showed that GSK3 functioned to inhibit the uptake of another axon repulsion cue Slit receptor Robo1, and deleting endophilin led to reduced axon growth of cortical neurons. Lastly, it was shown that GSK3 and Cdk5 also regulate FEME carrier budding from the membrane by inhibiting microtubule-based motor dynein and GTPase dynamin. Overall, the study identified Cdk5 and GSK3 as important kinases regulating FEME at two key steps, revealing a novel regulatory mechanism of FEME. The cell biological and biochemical experiments were performed nicely with clear results. There are still some major concerns that need to be addressed, especially the functional experiments using cultured PC12 cells and rat cortical neurons. In addition, the manuscript should provide more detailed description about the experimental conditions and result interpretations for better understanding.

1. In Figure 1, the identification of Cdk5 and GSK3 was based on small molecule inhibitors, which usually have non-specific effects. Because it was the main conclusion of the study and the rest of the result interpretation also depended on it, it is worthwhile to further confirm the pharmacological results with clear genetic approaches. For instance, siRNAs against Cdk5 or GSK3s and rescuing plasmids encoding siRNA-resistant Cdk5 or GSK3s could be used to provide unequivocal evidence. Dominant negative mutants of each kinases could also be used.

In Figure 1D, to provide direct evidence that serum-induced FEME activation was due to GSK3 inactivation, a constitutive active mutant of GSK3b (S9A) could be used to see if it could antagonize the serum effect on FEME.

2. In Figure 2H, mutant CRMP4 that cannot bind to endophilin led to enhanced neurite outgrowth from NGF treated PC12 cells. The reason was explained to be due to increased levels of semaphorin receptor Plexin A1 at the cell surface. However, there was no direct experimental evidence supporting the conclusion. Was there Sema3A in the culture medium? Would knocking down Plexin A1 abolish the increased neurite outgrowth?

Moreover, deleting endophilin in rat cortical neurons (shown in Figure 3D) led to reduced axon growth, due to increased Robo1 on the neuronal surface. Such results seemed confusing when compared with the PC12 results.

3. In Figure 3, better images of rat cortical neurons should be used. In addition, it is unclear how axon lengths were measured and quantified, and if knocking down endophilins affected neuronal polarization. It is also not clear if Slit1 was present in the culture. To directly show that decreased internalization of Robo1 was responsible for the reduced axon growth, knocking down Robo1 would be a good approach to support the conclusion.

4. In Figure 4, the interpretation of the results should be more cautious. For instance, there was no direct evidence supporting the conclusion that dynein acted to generate membrane tension and facilitate FEME carrier budding.

To all reviewers:

We thank all three reviewers for their helpful comments and suggestions. Following their advices, we considerably revised, and in our opinion improved, the manuscript. We updated **4 panels** with additional data (Figure 1b, 1c, 2a, 6a) and **added 36 totally new panels** (new Figure 2b, 2c, 2d, 2e, 2f, 2g, 3a, 3b, 3c, 3d, 4b, 5g, 6c, 6d, 6e, 7a, 7b, 7c, 7d, 7e, S1a, S1b, S1c, S2a, S2b, S3a, S3b, S3c, S3e, S3f, S4b, S6a, S6b, S6c, S6d and S6e).

The major findings added to the paper are the following:

- 1) we found that GSK3 β binds to, and is recruited by Endophilin on priming spots, but not FEME carriers (**new Figure 2**).
- 2) we established Cdk5 and GSK3 β act in synergy to inhibit FEME (**new Figure 3**).
- 3) we found that Bin1 is a FEME component and function by recruiting Dynein on FEME carriers. This function is opposed by Cdk5 and GSK3 β (**new Figure 6 & 7**).

We also removed the 3 panels from the original submission related to axon growth or cell collapse because we could not get better evidence (and we removed the corresponding claims). Therefore, the revised version is focused on the role of Cdk5 and GSK3 β on FEME and endocytosis of cargoes.

Details and evidence are presented in the manuscript and changes are explained in the response to each reviewer below:

Reviewer #1:

1. The authors observed that some cell types display robust FEME while others do not. If Cdk5 and GSK3 indeed play critical roles in controlling FEME activity, as supporting evidence, the authors should compare the activity levels of Cdk5 and GSK3 in cell lines growing in normal medium with 10% serum, e.g. HUVEC and RPE1 vs HeLa, HEK293 or BSC1.

> We thank the reviewer for the suggestion. We performed the experiments for GSK3 β , for which an antibody exists to measure its inactive state (phosphorylated Ser 9 GSK3 β) and the results are shown on the new Figure 2. We found a good ($r^2=0.99$) correlation between GSK3 β activity and the number of FEME carriers produced upon serum starvation, resting state and extra serum addition: the less phosphorylated Ser 9 GSK3 β there was (*ie*, the more GSK3 β was active), the less FEME was active (**new Fig. 2a-b**). However, acute stimulation of β 1 adrenergic receptor (with dobutamine) could increase FEME activity without detectable changes in GSK3 β activity (**new Fig. 2a-b**).

When we measured GSK3 β activity in 6 cell lines and primary cells (HeLa, HEK293, BSC1, RPE1, human dermal primary fibroblasts hDFA, and HUVEC) we found a good ($r^2=0.91$) correlation between GSK3 β activity and the percentage of cells (in resting populations) that were displaying spontaneous FEME. Surprisingly the correlation was inverted: the more GSK3 β was active, the most spontaneous FEME we observed (**new Fig. 2e-f**). However, the correlation was not significant when we measured the number of FEME carriers produced (**new Fig. 2g**). Yet, extra serum addition stimulated FEME beyond resting levels in all cells (**new Supplementary Figure 1c**). Thus, we concluded that GSK3 β activity on its own was not a predictor of FEME activity (because the activity of the priming kinases and/or of antagonistic phosphatases might be dominant). But, for all cells tested, reducing GSK3 β activity (by adding FBS) was sufficient to promptly activate FEME.

As we detected GSK3 β binding to Endophilin and GSK3 β recruitment to priming Endophilin spots at the leading edge, but not on budded FEME carriers (**new Fig. 2c-d**), we concluded that GSK3 β was a major regulator of FEME.

Unfortunately, we could not do the same experiments with Cdk5, as we did not find reliable antibodies that would be specific for Cdk5 phosphorylation and work by immunostaining (to correlate single cell levels and FEME activity). We tried antibodies against phosphorylated Ser 778 Dynammin-1, phosphorylated Ser 522 CRMP4 and phosphorylated Ser 732 FAK1, but none of them worked well by immunostaining.

Nevertheless, as we had good anti-Cdk5 antibodies, we could confirm the involvement of Cdk5 by using RNAi-mediated depletion of the kinase (**new Fig. 3a-d and new Supplementary Figure 2b**). There, as for GSK3 β , total cellular levels of Cdk5 correlated well ($r^2=0.97$) with FEME: the less Cdk5 or GSK3 β the cells had, the more spontaneous FEME they did (same cell measurements). The effect could be rescued by wild-type or constitutively active, but not dominant-negative, kinases (**new Fig. 3a and c**).

Together, we believe that we strengthened the conclusion that Cdk5 and GSK3 β control FEME activity.

2. Figure 2E-G: It was reported that Semaphorin 3A-induced receptor endocytosis is mediated by L1 CAM and is AP2-dependent (Castellani V et al., Mol Cell Neurosci 26: 89-100, 2014), whereas FEME is AP2-independent. To determine whether or not it is endophilin-dependent and clathrin-independent, the authors need to test the effect of endophilin depletion and CME inhibition on internalization of Plexin A1 upon semaphoring 3A treatment.

> This is a good point. AP2 depletion induced a ~75% reduction in total PlexinA1 uptake, whereas Endophilin A1+A2+A3 triple knocked-down (TKD) cells had a stronger inhibition (>90%) (**new Supplementary Figure 3e**). Consistently, we could detect fed anti-PlexinA1 antibodies in as many (~30%) of FEME carriers in control or AP2 KD cells (**new Figure 5g**). We amended the text accordingly. We concluded that, as other receptors (e.g. EGFR), PlexinA1 can enter into cells through both CME and FEME. However, we noted that FEME was still active in AP2KD cells, but CME could not compensate for Endophilin TKD.

3. Figure 2G: the authors claim that the longer neurite phenotypes in Figure 2H and cell collapse in Figure 2I are likely mediated by the increased levels of Plexin A1, caused by FEME inhibition or uncoupling of CRMP4 to endophilin. If it is true, there should be more plexin A1 signals on the cell surface (detect either by immunostaining or surface protein biotinylation) in EEN TKO cells or cells overexpressing CRMP4 mutants. Rather than speculating, the authors should provide experimental data to prove it.

> Despite several attempts, we could not get improve our data - thus, we removed neurite extension and cell collapse data and corresponding claim from the paper. Instead, we limited our study to endocytosis of PlexinA1 and strengthened the paper on the role of Cdk5 and GSK3 β .

4. In Figure 2H, the authors used neurite outgrowth of PC12 cells as readout to assess the role of Plexin A1 uptake in FEME carriers, and found that overexpression of endophilin-binding deficient mutants of CRMP4 caused increase in neurite length. However, as it is well known that NGF is the major growth factor that induces neurite outgrowth of PC12, the authors should either use a cellular model for semaphoring 3A signaling or clarify whether the phenotype is caused by defect in plexin A1 uptake or in endophilin-mediated FEME of the NGF-TrKA signal complex.

> As above, we could not get conclusive data and removed the claim from the paper.

5. Figure 3B: the experimental group of 2 nM Slit1-treated cells is missing.

> While we were revising this manuscript, the Eichmann lab published a nice story reporting that VEGFR2, Slit2 and ROBO1 enter through Endophilin-mediated endocytosis (Genet et al. Nat Commun 2018). They established that the BAR domain protein srGAP1 bridged Endophilin and ROBO1. Thus, we decided that we did not have much to add than confirming that the Endophilin-mediated endocytosis they reported was indeed FEME and showed that Slit1 was indeed internalized into FEME carriers, and, consistent with our other findings, inhibiting Cdk5 and GSK3 β increased the uptake (**new Supplementary Figure 4b**). We cited Genet et al. and toned down the claims accordingly in the text and abstract.

6. Figure 3C: 1) The signal intensity of Slit1 in AP2 KD cells looks much weaker than control, which is inconsistent with the quantification results in the right panel. How many cells were analyzed? 2) In figure legends the authors said it is "Fluorescently labeled Slit1 uptake", whereas in METHODS it was described as "...cells were stained with antibodies or phalloidin...". Please clarify which labeling method was used in the Slit uptake assay. 3) If GSK3 indeed inhibits Slit-induced Robo uptake, treatment of cells with GSK3 inhibitor should also cause an increase in Slit1 uptake that is opposite to the phenotype caused by Endo TKD.

> We are sorry for the mistake and confusion: Slit1, which had a (His)6 tag, was added to live cells at 37C. Internalized Slit1 was detected with an anti-His tag antibody post fixation. We have redone the whole experiment (**new Supplementary Figure 4b**), including additional conditions (Cdk5i+GSK3i, CDK5+GSK3 TKD, etc.) and clarified the legends and the Methods.

7. Figure 3D: again there is inconsistency in the figure legend and METHODS. The cells were cortical neurons according to the legend. In METHODS they were described as hippocampal neurons. Furthermore, did the authors determine the KD efficiency of EEN1, 2 and 3 in neurons? To measure axon extension, the authors should take pictures of whole cells and measure the length of axons of individual neurons (at least 30 neurons/group), not Tau total signal intensity of the view field, which most likely represents cell density, rather than the extent of axon outgrowth.

> We tried both but as could not get better data, we removed the claim from the paper.

8. Figure 4. The data showing that kinesin is not involved in betaAR1 uptake is very nice. However, the experiments designed to determine whether the dynein motor complex and microtubules are involved in FEME carrier budding and corresponding conclusions are similar to those reported by Renard et al. in their study of endophilin A2-mediated Shiga Toxin B endocytosis (Renard H-F et al., Nature 517, 493-

496, 2015). It would be better to say these data (in Figure 4-D) are in agreement with previous study and cite the paper.

> Absolutely. We did not mean to sound as we discovered it. We moved the data to supplement (now **Figure S5**) and amended the text accordingly, citing the relevant papers.

9. The authors showed that inhibition of Cdk5 and GSK3 increased localization of dynein to endophilin-labeled structures, and stated “this is consistent with the regulation of cargo-binding complexes loading onto Dynein by Cdk5 and GSK3 kinases”. However, it was reported that Cdk5 phosphorylation of the dynein regulator Ndel1 stimulates cargo transport by promoting dynein-microtubule interaction (J Neurosci 31, 17207-17219, 2011 and Traffic 18, 808-824, 2017). The increased colocalization between the motor and cargo carriers might be a result of failure of the motor to move on microtubule tracks, hence the accumulation of dynein-associated EPAs rather than a cargo loading defect. Therefore, the authors should interpret the data more carefully to explain this apparent discrepancy. A motor-cargo interaction assay (e.g. vesicle IP using antibodies against endophilin or the transmembrane receptor cargoes) is a good approach to determine whether or not the kinases regulate cargo loading onto dynein. It would also help if the authors test whether kinase inhibition affects the distribution of FEME carriers to microtubules by immunostaining (similar to Figure 4D).

> This was a very good comment, thank you. We detected Dynein on immuno-isolated FEME carriers purified from cells stimulated with FBS (a natural way to decrease GSK3 β activity) (new **Figure 6d-e**), and more Dynein was immunoprecipitated from cells treated with Cdk5i (new **Figure 6c**). And the increased Dynein presence on FEME carriers (**Figure 6b**) induced by Cdk5i+GSK3i was accompanied increase budding rate (**Figure 3e**) and β 1AR, PlexinA1 and Slit1 uptake (**Figure 1d, 5g and S4b**). In addition we now identified that Bin1, not Endophilin, binds to Dynein and recruits it on FEME carriers (new **Figure 7**). However, we could not rule out that Cdk5 and/or GSK3 regulate Dynein processivity as well (as reported in the literature), and we amended the text accordingly.

Minorpoints:

1. Figure 2E, top left label: “anti-Endophilin” should be “merged” or “Endo EGFP-CRMP”?

> We fixed it, thank you (now Fig. 5e)

2. Supplementary Figure S1C, top left label “anti-Endophilin” should be “merged” or “Endo EGFP”?

> We fixed it, thank you (now Fig. 5e)

Similarly, in panel D, the images labeled with “EndophilinA2-RFP” are merged images of RFP and GFP.

> We fixed it, thank you (now Fig. S3d)

3. Supplementary Figure S1D, how was colocalization quantified? Pearson coefficient or Mander's overlap?

> It was Manders' overlap, we specified it in the Methods.

4. In “QUANTIFICATION AND STATISTICAL ANALYSIS”, the authors stated that “P values < 0.01 were considered significant”, whereas in Figure 1B, *p < 0.05 is considered significant. Please be consistent.

> Sorry for the carelessness. We fixed it.

Reviewer#3:

The major concern I have is the sole use of endophilin immunofluorescence signals as readout for the FEME pathway. Endophilin does participate in clathrin-mediated endocytosis (CME). The effect of these chemical inhibitors could be pleiotropic because many of their substrates also work in CME. Lastly, the recruitment of endophilin to the plasma membrane and its duration on the membrane could be independent from the actual uptake (For instance, the amount of clathrin accumulated on the membrane is most likely not an accurate readout for the efficiency of CME). Therefore, it is rather surprising that the authors did not provide evidence that these inhibitors do affect the uptake of the receptors, either positively or negatively, and whether such effects are clathrin-independent.

> We agree and we now have direct evidence for β 1AR (new **Figure 1d**), PlexinA1 (new **Figure 5g**) and ROBO1 (through uptake of its ligand Slit1, new **Supplementary Figure S4b**)

For example, line 112, “Increased levels of Plexin A1 at the cell surface”: was it shown?

> We could not get convincing measurements of cell surface levels of PlexinA1 so we removed the speculation from the paper.

Does the number of endophilin dot positively correlate with endocytosis of these receptors?

> Yes it does, please see new Figure 1, 5 and S4: inhibition of Cdk5 and GSK3 β (both using inhibitors and RNAi-mediated depletion) increased FEME carriers (endophilin spots in the cytoplasm) that contained β 1AR, PlexinA1 or Slit1 in them.

In addition, how the scoring system worked was somewhat obscure. Was it based on automatic quantitation or visual impressions? It was not mentioned in the method.

> We clarified it both in the text, Figure legend and Methods.

Other comments:

1. The recruitment of EGFP-tagged CRMP4 or R525E onto FEME carriers in Fig 2 is rather difficult to appreciate. Also the data showing the increased colocalization upon Cdk5 inhibitor treatment requires some faith to believe in.

> As we could not get better data (both EGFP-CRMP4 and anti-CRMP4 antibodies we tested gave some diffuse background), we complemented and improved the PlexinA1 uptake (using an antibody recognizing the ectodomain of PlexinA1). We also added RNAi experiments, to confirm the involvement of CRMP4 in PlexinA1 uptake by FEME (**new Figure 5g and Supplementary Figure S3e-f**).

2. Do endophilin and ROBO1 really colocalize? The two signals do not seem to be in the same compartments.

> We could not get better data as the anti-ROBO1 antibodies we used gave some background, so we removed the experiment from the paper. Instead, we used recombinant Slit1, which had a (His)6 tag. Internalized Slit1 was detected with an anti-His tag antibody post fixation. This approach gave much better signal to noise data, and the colocalization of Slit1 into FEME carriers was convincing (**new Supplementary Figure 4b**).

3. The involvement of CRMP4 was tested by biochemical experiments and dominant-negative mutants. To demonstrate that it is the key substrate of Cdk5 that affects FEME, some loss-of-function experiments are needed.

> We agree and we added it to the paper (**new Figure 5g and new Supplementary Figure 5e-f**). It confirmed the involvement of CRMP4 in the uptake of PlexinA1 into FEME carriers.

4. Fig 3c needs to show the GSK3 inhibitor under these conditions. AP2 KD seems to reduce Slit1 internalization in the image shown, which seems to contradict with the statement in the text.

> We agree and we added it to the paper. As explained above (point 2) we redid the experiment entirely using recombinant Slit1 (**new Supplementary Figure 4b**). We included both inhibitors and RNAi of the kinases to confirm their involvement.

5. How does GSK regulate the sorting of ROBO1?

> We tried few hypotheses but unfortunately did not get conclusive evidence. While we were revising this manuscript, the Eichmann lab published a nice story reporting that VEGFR2, Slit2 and ROBO1 enter through Endophilin-mediated endocytosis (Genet *et al. Nat Commun* 2018). They established that the BAR domain protein srGAP1 bridged Endophilin and ROBO1. GSK3 β might act on srGAP1 (instead of on ROBO1 or Endophilin) but we did not test it.

We cited Genet *et al.*, moved our ROBO1/Slit1 data in supplement (**new Supplementary Figure 4**) and updated the claims accordingly in the text and abstract.

6. Line 134 and the following paragraph, does the author mean “fission” instead of “budding”?

> Yes indeed. We rephrase it in the revised version.

7. Are dynein recruited to the late stage of FEME similar to Fig 1F?

> We don't know. We could not get EGFP-tagged Dynein construct to work in our hands.

8. Line 80 has unknown characters.

> Thank you, we fixed it.

9. Line 108 has missing reference.

> Thank you, we fixed it.

Reviewer #4:

There are still some major concerns that need to be addressed, especially the functional experiments using cultured PC12 cells and rat cortical neurons.

> We agree. As we could not get better evidence, we removed these experiments and claim from the paper. Instead, we focused the paper on endocytosis and strengthened the case that Cdk5 and GSK3 β inhibit FEME (see revised manuscript).

In addition, the manuscript should provide more detailed description about the experimental conditions and result interpretations for better understanding.

> We agree and we improved the manuscript and the method.

1. In Figure 1, the identification of Cdk5 and GSK3 was based on small molecule inhibitors, which usually have non-specific effects. Because it was the main conclusion of the study and the rest of the result interpretation also depended on it, it is worthwhile to further confirm the pharmacological results with clear genetic approaches. For instance, siRNAs against Cdk5 or GSK3s and rescuing plasmids encoding siRNA-resistant Cdk5 or GSK3s could be used to provide unequivocal evidence. Dominant negative mutants of each kinases could also be used.

In Figure 1D, to provide direct evidence that serum-induced FEME activation was due to GSK3 inactivation, a constitutive active mutant of GSK3b (S9A) could be used to see if it could antagonize the serum effect on FEME.

> Thank you for the suggestion. We did as advised and the data strengthened our conclusions (**new Figure 3a-d and new Supplementary Figure 2b**)

2. In Figure 2H, mutant CRMP4 that cannot bind to endophilin led to enhanced neurite outgrowth from NGF treated PC12 cells. The reason was explained to be due to increased levels of semaphorin receptor Plexin A1 at the cell surface. However, there was no direct experimental evidence supporting the conclusion. Was there Sema3A in the culture medium? Would knocking down Plexin A1 abolish the increased neurite outgrowth?

> Despite several attempts, we could not improve our data - thus, we removed neurite extension and the corresponding claims from the paper. Instead, we limited our study to the endocytosis of PlexinA1 and refocused the paper on Cdk5 and GSK3 β .

Moreover, deleting endophilin in rat cortical neurons (shown in Figure 3D) led to reduced axon growth, due to increased Robo1 on the neuronal surface. Such results seemed confusing when compared with the PC12 results.

> As above, we could not improve our data and we removed it from the paper.

3. In Figure 3, better images of rat cortical neurons should be used. In addition, it is unclear how axon lengths were measured and quantified, and if knocking down endophilins affected neuronal polarization. It is also not clear if Slit1 was present in the culture. To directly show that decreased internalization of Robo1 was responsible for the reduced axon growth, knocking down Robo1 would be a good approach to support the conclusion.

> As above, we could not improve our data and we removed it from the paper. Instead we provided new data to strengthen our conclusions on the kinases (please see the revised manuscript):

1) we found that GSK3 β binds to, and is recruited by Endophilin on priming spots, but not FEME carriers (**new Figure 2**)

2) we established Cdk5 and GSK3 β act in synergy to inhibit FEME (**new Figure 3**)

3) we found that Bin1 is a FEME component and function by recruiting Dynein on FEME carriers. This function is opposed by Cdk5 and GSK3 β (**new Figure 6 & 7**)

4. In Figure 4, the interpretation of the results should be more cautious. For instance, there was no direct evidence supporting the conclusion that dynein acted to generate membrane tension and facilitate FEME carrier budding.

> We agree and we revised the text accordingly.

Reviewers' comments:

Reviewer #1 (Remarks to the Author):

The authors have addressed all of my concerns about the original manuscript. I do have some minor concerns for the authors to address before publication.

1. Fig. 5e: the S522A mutation enhanced binding of CRMP4 to endophilin as shown by colP. However, localization of the mutant to EPA is lower than WT in immunostaining experiments. The authors should provide explanation for the inconsistency.
2. Fig. 5f: the authors should provide IF images for the + GSK3i and +Cdk5i+GSK3i experimental groups.
3. Fig. 6e: 1) How did the authors probe lipids in the immunoisolation experiments? Please provide information about antibodies used for lipids. 2) There was no signal for endophilin in the input lane of the immunoblot, authors should replace the blot with a better one to justify the quantitative data of the right panel.
4. Fig. 7e. Please provide confocal image data for Bin1-Dynein in resting and +Cdk5i+GSKi-treated cells.
5. There are quite a few errors in the text (see below for examples), the authors should do a thorough proofreading before submission of the final version.

Line 46, p2 : "at his stage"

Line 38, p14: 10 mM dobutamine?

Line 45, p14: spots within 1 mm of cell edges

Reviewer #3 (Remarks to the Author):

Unfortunately, the authors did not address my major concerns. The removal of functional data also compromised the significance of the work. I am sorry that I could not be more positive in this situation.

Reviewer #4 (Remarks to the Author):

The manuscript has been significantly revised by removing part of the results regarding axon growth. New experiments using siRNAs against CDK5 and GSK3s, as well as the rescuing experiments, were performed. The new data shown in Fig. 3 greatly strengthened the conclusion that CDK5 and GSK3 acted to regulate FEME. Because several of the comments raised were regarding neurons, which were all removed in the revised manuscript, the new manuscript now mainly focused on the regulation of FEME and endocytosis by GSK3 and CDK5. Therefore, it seems that one potential weakness is the lack of physiological functional consequences about the newly identified regulatory mechanism. Some discussion regarding the issue would improve the manuscript.

Point by point response

Reviewer #1:

The authors have addressed all of my concerns about the original manuscript. I do have some minor concerns for the authors to address before publication.

We thank the reviewer for their support.

1. Fig. 5e: the S522A mutation enhanced binding of CRMP4 to endophilin as shown by coIP. However, localization of the mutant to EPA is lower than WT in immunostaining experiments. The authors should provide explanation for the inconsistency.

Thank you for noticing the issue. This was an unfortunate mistake in the statistical symbols on the graph. The colocalization of S522A is actually not statistically different from WT (see Source data we provided with the previous submission).

We are sorry for the mistake and corrected the symbols on Fig. 5e:

2. Fig. 5f: the authors should provide IF images for the + GSK3i and +Cdk5i+GSK3i experimental groups.

We supplied the image (on Suppl Fig. 3e to avoid crowding Figure 5).

3. Fig. 6e: 1) How did the authors probe lipids in the immunisolation experiments? Please provide information about antibodies used for lipids.

We actually did not use an antibody but alcohol-free coomassie staining, which we discovered to quantitatively stain lipids (Boucrot et al. Cell 2012). We had the reference in the Methods (page 14), but we have now clarified the text and figure legend as well (highlight in yellow on page 5 and 14).

2) There was no signal for endophilin in the input lane of the immunoblot, authors should replace the blot wither better one to justify the quantitative data of the right panel.

This is a good point, we added a high exposure blot with the input visible to Fig. 6e:

4. Fig. 7e. Please provide confocal image data for Bin1-Dynein in resting and +Cdk5i+GSKi-treated cells.

We supplid the image (on Suppl Fig. 6e to avoid crowding Figure 7).

5. There are quite a few errors in the text (see below for examples), the authors should do a thorough proofreading before submission of the final version.

Line 46, p2 : "at his stage"

Line 38, p14: 10 mM dobutamine?

Line 45, p14: spots within 1 mm of cell edges

Apologies for the typos, we proof-read the manuscript and fixed the typos.

Reviewer #3:

Unfortunately, the authors did not address my major concerns. The removal of functional data also compromised the significance of the work. I am sorry that I could not be more positive in this situation.

We regret that, even though we addressed 11 out of the 13 points raised by the reviewer, we did not manage to convince them.

However, we have now re-introduce some functional data into the manuscript. After changing experimental conditions (mouse instead of rat neurons and a layer of feeder cells to support electroporated neurons), we managed to confirm the main conclusions in mouse hippocampal neurons:

1. The sorting of PlexinA1 into FEME carriers (through the binding of Endophilin to the Plexin A1 adaptor CRMP4) was required for axon elongation and branching. This is because it prevents premature axon growth cone collapse.
2. The phosphorylation of CRMP4 by Cdk5, which antagonizes the binding to Endophilin, was required for proper neuron axon elongation and branching.

The new data is in new Fig. 5h-j and Suppl. 3h-l and highlighted in yellow in the text

Reviewer #4:

The manuscript has been significantly revised by removing part of the results regarding axon growth. New experiments using siRNAs against CDK5 and GSK3s, as well as the rescuing experiments, were performed. The new data shown in Fig. 3 greatly strengthened the conclusion that CDK5 and GSK3 acted to regulate FEME.

We thank the reviewer for their support.

Because several of the comments raised were regarding neurons, which were all removed in the revised manuscript, the new manuscript now mainly focused on the regulation of FEME and endocytosis by GSK3 and CDK5. Therefore, it seems that one potential weakness is the lack of physiological functional consequences about the newly identified regulatory mechanism. Some discussion regarding the issue would improve the manuscript.

However, we have now re-introduce some functional data into the manuscript. After changing experimental conditions (mouse instead of rat neurons and a layer of feeder cells to support electroporated neurons), we managed to confirm the main conclusions in mouse hippocampal neurons:

1. The sorting of PlexinA1 into FEME carriers (through the binding of Endophilin to the Plexin A1 adaptor CRMP4) was required for axon elongation and branching. This is because it prevents premature axon growth cone collapse.

2. The phosphorylation of CRMP4 by Cdk5, which antagonizes the binding to Endophilin, was required for proper neuron axon elongation and branching.

The new data is in new Fig. 5h-j and Suppl. 3h-l and highlighted in yellow in the text

We also discussed the findings in the Discussion (highlighted in yellow), as suggested.

REVIEWER COMMENTS

Reviewer #1 (Remarks to the Author):

In the first revision, the authors have provided evidence that 1. In addition to small molecule inhibitors, Cdk5 and GSK3beta knockdown inhibit FEME; 2. Cdk5 and GSK3beta control the uptake of FEME cargoes beta1AR, Plexin A1 (via CRMP4) and Slit1-ROBO1 (via ROBO1) by showing changes in colocalization of the receptors/ligands with endophilin-labeled structures (EPA). In addition, the authors also show that uptake of Plexin A1 I requires both CME and FEME. The manuscript has been further improved after the second round revision by specifically addressing the issue about physiological significance of Cdk5 and GSK3beta-regulated, endophilin-mediated Plexin A1 internalization with axon outgrowth data from mouse hippocampal neurons (instead of PC12 cells). However, I do have some minor concerns for the authors to address before publication.

1. Line 107-109 and Figure 1d:

1) What do LHS and RHS stand for in the right panel of Figure 1d?

2) As the authors claimed that inhibition of Cdk5 or GSK3 increased productive FEME (beta1AR and endophilin double positive structures), is there any statistically significant difference between the control and +Cdk5i or +GSK3i experimental groups?

3) Further, the authors stated that "This is also coherent with competition between G proteins and Endophilin to bind to beat1AR". This sentence is confusing. The authors should explain what it means.

2. Figure 2f: check the label below the X axis.

3. Figure 3d-e: does the histogram in Figure 3e show frequency of endophilin-labeled FEME carriers? It is hard to tell whether this histogram represents quantification results of data shown in Figure 3d or kymographs from +Cdk5i and +GSK3i cells. The authors must clarify to which imaging data the histogram corresponds to.

4. Figure 4d: the authors should add confocal imaging data from resting cells and single inhibitor treated cells in the left panel to match with quantification data in the right panel.

5. Figure 4g and Suppl Fig. 4c: if "resting" (versus stimulated or induced) means no ligand (semaphorin or slit), the authors should change the labeling for the + Sema 3A or + slit experimental groups in both image and histogram panels to avoid confusion.

6. Figure 6b: again, representative confocal images for all 5 experimental groups should be displayed to match with the histogram.

7. Please explain why disruption of CRMP4-Endophilin interaction caused reduction of axon branching and constitutive growth cone collapse even in the absence of semaphorin 3A.

8. Line 198: To help people outside the neurobiology field understand the part about axon outgrowth, the authors should briefly explain mechanism for Semaphorin 3A-plexin A1-mediated growth cone collapse.

9. There are still quite a few errors in the text and figures. The authors should proofread carefully before submission of the final version. For example,

Line 83-84: Small molecule inhibitors were used to screen the role of kinases known to regulate membrane trafficking and actin cytoskeleton dynamics (which is required for FEME) was performed using small molecule inhibitors.

Line 209: increased

Line 211: thereby regulating the, uptake of Plexin A1

Line 252: discreet or discrete?

Reviewer #4 (Remarks to the Author):

In the revised manuscript, new functional data regarding hippocampal neuron axon growth, branching, and growth cone collapse were added, demonstrating physiological significance of the identified pathway and regulatory mechanism. Below are some minor comments for the new data.

1. In Fig. 5h, it would be much clearer and more convincing if the data is presented as average axon lengths from each independent experiment, rather than individual neurons from all experiments.
2. In Fig. 5i, the phenotype of axon branching was very strong. It is not clear what is the underlying cellular mechanism. Previous studies indicated that culture hippocampal neuron axons form branches mainly via interstitial branching. Did expression of R525E mutant of CRMP4 affect such process? If so, how? Some discussion would help the readers better understand the observe results. Similarly, the data presentation and statistics should be clearly presented in detail.
3. In Fig. 5j, the growth cone collapse in the absence of SemaA3 when CRMP4 R525E was expressed was very interesting. In the discussion it was explained to be induced by endogenous secreted Sema3A. Could it be induced by the direct effect of mutant CRMP4 in the growth cones?

Reviewer #1

In the first revision, the authors have provided evidence that 1. In addition to small molecule inhibitors, Cdk5 and GSK3beta knockdown inhibit FEME; 2. Cdk5 and GSK3beta control the uptake of FEME cargoes beta1AR, Plexin A1 (via CRMP4) and Slit1-ROBO1 (via ROBO1) by showing changes in colocalization of the receptors/ligands with endophilin-labeled structures (EPA). In addition, the authors also show that uptake of Plexin A1 I requires both CME and FEME. The manuscript has been further improved after the second round revision by specifically addressing the issue about physiological significance of Cdk5 and GSK3beta-regulated, endophilin-mediated Plexin A1 internalization with axon outgrowth data from mouse hippocampal neurons (instead of PC12 cells). However, I do have some minor concerns for the authors to address before publication.

We thank the reviewer for the feedback and advices to improve the manuscript.

1. Line 107-109 and Figure 1d:

1) What do LHS and RHS stand for in the right panel of Figure 1d?

> It stands for Left Hand Side (LHS) and Right Hand Side (RHS), referring to the two Y-axes on the left and right sides of the graph on Figure 1d. We clarified the figure legend. We combined two graphs into a grouped column chart to keep the figure balanced and succinct.

2) As the authors claimed that inhibition of Cdk5 or GSK3 increased productive FEME (beta1AR and endophilin double positive structures), is there any statistically significant difference between the control and +Cdk5i or +GSK3i experimental groups?

> This was a typo error, it should read 'and' not 'or': as only the dual inhibition of Cdk5 and GSK3 β increase significantly β 1AR uptake. We corrected the text and clarified Fig 1d by adding the statistics on the +Cdk5i and +GSK3i single inhibitions.

3) Further, the authors stated that "This is also coherent with competition between G proteins and Endophilin to bind to beta1AR". This sentence is confusing. The authors should explain what it means.

> We apologise for the confusion: we rephrased to: '[...] the overlap of the binding sites of G proteins and Endophilin on β 1-AR'.

2. Figure 2f: check the label below the X axis.

> Thank you, we fixed the label.

3. Figure 3d-e: does the histogram in Figure 3e show frequency of endophilin-labeled FEME carriers? It is hard to tell whether this histogram represents quantification results of data shown in Figure 3d or kymographs from +Cdk5i and +GSK3i cells. The authors must clarify to which imaging data the histogram corresponds to.

> Sorry for the confusion. Yes the histograms related to the kymographs from cells treated with the inhibitors (Figure 3e) - we updated the labels for clarity.

4. Figure 4d: the authors should add confocal imaging data from resting cells and single inhibitor treated cells in the left panel to match with quantification data in the right panel.

> We added the images to Figure 4d.

5. Figure 4g and Suppl Fig. 4c: if "resting" (versus stimulated or induced) means no ligand (semaphorin or slit), the authors should change the labeling for the + Sema 3A or + slit experimental groups in both image and histogram panels to avoid confusion.

> We trust the reviewer meant Figure 5g instead of 4g (Figure 4 only has 4 panels). We reserved the 'resting' label for unperturbed cells and changed the other 'resting' labels for 'control' on Fig. 5g and Suppl. Fig. 3g and Suppl Fig. 4c for clarity.

6. Figure 6b: again, representative confocal images for all 5 experimental groups should be displayed to match with the histogram.

> We added the images and moved the panel to Figure 6d as we rearranged the Figure to fit the new images.

7. Please explain why disruption of CRMP4-Endophilin interaction caused reduction of axon branching and constitutive growth cone collapse even in the absence of semaphorin 3A.

> This is an important point. The disruption of CRMP4-Endophilin interaction prevented Plexin A1 from being internalized by FEME (Fig. 5e-g). Even though we could not measure it directly, we expect that PlexinA1 accumulated at the cell surface following the disruption. We previously measured that FEME cargoes of which internalization is blocked accumulate at the cell surface, including β 1-adrenergic and

α 2a-adrenergic receptors, dopaminergic D3 and D4 receptors, muscarinic acetylcholine receptor 4 as well as epidermal growth factor (EGFR) receptors (Boucrot *et al.* 2015, PMID25517094).

As such, neurons overexpressing the mutants and with increased cell surface Plexin A1 levels likely became sensitive to the low endogenous levels of Semaphorin 3A secreted into the media by neuronal cultures (Wang *et al.* 2018, PMID30353093). Semaphorin 3A, through its action on Plexin A1, is a potent inducer of axon growth cone (AGC) collapse, and thus inhibits axon branching and elongation (Dent *et al.* 2004, PMID15044539). As both axon branching and elongation depends on properly formed AGC (Kali and Dent 2014, PMID24356070), increased Plexin A1 plasma membrane levels is expected to make neurons hypersensitive to Semaphorin 3A and therefore to induce constitutive AGC collapse and decreased axon branching and elongation.

We added this explanation to the text to clarify this point.

To test this further, we also quantified branching and found that overexpression of the CRMP4 R525E mutant that cannot bind to Endophilin significantly decreased interstitial branching (added to Fig. 5i). This confirmed the reduced general branching and decreased axon length, and increased axon growth cone collapse caused by the mutant we already reported:

8. Line 198: To help people outside the neurobiology field understand the part about axon outgrowth, the authors should briefly explain mechanism for Semaphorin 3A-plexin A1-mediated growth cone collapse.

> We updated the text to clarify the mechanism.

9. There are still quite a few errors in the text and figures. The authors should proofread carefully before submission of the final version.

> Thank you, we proofread the text carefully and caught several typos.

For example,

Line 83-84: Small molecule inhibitors were used to screen the role of kinases known to regulate membrane trafficking and actin cytoskeleton dynamics (which is required for FEME) was performed using small molecule inhibitors.

Line 209: increased

Line 211: thereby regulating the, uptake of Plexin A1

Line 252: discreet or discrete?

> We fixed all the above.

Reviewer #4

In the revised manuscript, new functional data regarding hippocampal neuron axon growth, branching, and growth cone collapse were added, demonstrating physiological significance of the identified pathway and regulatory mechanism. Below are some minor comments for the new data.

We thank the reviewer for acknowledging the improvements and for the advices to improve the manuscript.

1. In Fig. 5h, it would be much clearer and more convincing if the data is presented as average axon lengths from each independent experiment, rather than individual neurons from all experiments.

> We have modified Fig 5h accordingly (see panel below).

2. In Fig. 5i, the phenotype of axon branching was very strong. It is not clear what is the underlying cellular mechanism. Previous studies indicated that culture hippocampal neuron axons form branches mainly via interstitial branching. Did expression of R525E mutant of CRMP4 affect such process?

> This was a very good suggestion. We quantified interstitial branching and found that overexpression of the R525E mutant not only reduced axon length and increased the proportion of axons without branching, it did so by significantly reduced interstitial branching. We added the data to Figure 5i:

This is consistent with the increased proportion of axon growth cones being collapsed in neurons overexpressing R525E as seen in Fig. 5j

If so, how? Some discussion would help the readers better understand the observe results.

>The disruption of CRMP4-Endophilin interaction upon the overexpression of the CRMP4 R525E mutant, prevented Plexin A1 from being internalized by FEME (Fig. 5e-g). Even though we could not measure it directly, we expect that PlexinA1 accumulated at the cell surface following the disruption. We previously measured that FEME cargoes of which internalization is blocked accumulate at the cell surface, including β 1-adrenergic and α 2a-adrenergic receptors, dopaminergic D3 and D4 receptors, muscarinic acetylcholine receptor 4 as well as epidermal growth factor (EGFR) receptors (Boucrot *et al.* 2015, PMID25517094).

As such, neurons overexpressing the mutants and with increased cell surface Plexin A1 levels likely became sensitive to the low endogenous levels of Semaphorin 3A secreted into the media by neuronal cultures (Wang *et al.* 2018, PMID30353093). Semaphorin 3A, through its action on Plexin A1, is a potent inducer of axon growth cone (AGC) collapse, and thus inhibits interstitial axon branching and elongation (Dent *et al.* 2004, PMID15044539). As both axon branching and elongation depends on properly formed AGC (Kali and Dent 2014, PMID24356070), increased Plexin A1 plasma membrane levels is expected to make neurons hypersensitive to Semaphorin 3A and therefore to induce constitutive AGC collapse and decreased axon branching and elongation.

We added this explanation to the text to clarify this point.

Similarly, the data presentation and statistics should be clearly presented in detail.

> We have updated the statistics on the Figure and presented all the raw data in the 'Source Data File'

3. In Fig. 5j, the growth cone collapse in the absence of SemaA3 when CRMP4 R525E was expressed was very interesting. In the discussion it was explained to be induced by endogenous secreted Sema3A. Could it be induced by the direct effect of mutant CRMP4 in the growth cones?

> We cannot rule out a direct effect of the CRMP4-R525E mutant, but the fact that the nearby S522A mutation did not give the same phenotypes suggested that the effect was mediated through the proline-

rich motif to which Endophilin binds to. Even though we do not know of another protein binding to this motif within CRMP4, we cannot rule it out.

In our view, the most likely explanation was that CRMP4-R525E mutant neurons with increased cell surface Plexin A1 levels became hypersensitive to the low endogenous levels of Semaphorin 3A secreted into the media by neuronal cultures (Wang et al. 2018, PMID30353093), as mentioned above. However, we added a sentence to the discussion to acknowledge the possibility of a direct effect of the CRMP4 mutant.